# CFA: Causality-Inspired Feature Augmentation for High-Dimensional Linear Regression

## Abstract

High-dimensional prediction with limited samples poses a significant challenge due to severe overfitting. While existing approaches tackle this via regularization, clustering, or representation learning, we introduce a novel framework inspired by causal inference that is designed to exploit latent structure linking predictors and responses. Our approach employs a new similarity-based clustering procedure guided by a metric that quantifies shared predictor–response dependencies, which tends to group variables that play similar roles with respect to (possibly latent) mediators or confounders. The resulting *causality-inspired* features are then incorporated into an augmented regression model, yielding sparser, more robust, and more generalizable predictions without attempting to recover the underlying causal graph. Experiments across synthetic and real-world datasets, including S&P 500 market data, demonstrate that our method achieves higher regression performance and markedly reduces overfitting compared to existing baselines.

## 1 Introduction

High-dimensional multivariate regression aims to predict a set of $n_2$ response variables $\mathbf{Y} \in \mathbb{R}^{N \times n_2}$ from $n_1$ predictors $\mathbf{X} \in \mathbb{R}^{N \times n_1}$ using $N$ samples. The standard formulation estimates a coefficient matrix $\mathbf{A} \in \mathbb{R}^{n_1 \times n_2}$ via ordinary least squares (OLS), $\widehat{\mathbf{A}}_{\text{OLS}} = \arg\min_{\mathbf{A}} \|\mathbf{Y} - \mathbf{X}\mathbf{A}\|_F^2$. This falls under the broader umbrella of Multi-Output Learning, which presents challenges in volume, variety, and veracity of output labels (Xu et al., 2019). A key challenge, especially in modern applications—from ecological modeling (Kocev et al., 2009) to stock selection in economics (Ghosn & Bengio, 1997)—is that the number of predictors and responses often exceeds the number of samples ($n_1, n_2 > N$), rendering OLS ill-posed and prone to overfitting (Hastie et al., 2015). In such regimes, the learned models tend to memorize the training data rather than uncover meaningful structure, leading to poor generalization. Many methods have been proposed to address this problem, most of which fall into three broad categories: regularization-based estimators, latent representation models, and clustering-based methods.

**Regularization-based methods** impose penalties on the coefficient matrix $\mathbf{A}$ to discourage undesired properties such as large values or complexity. Sparse regression techniques such as Lasso (Tibshirani, 1996) apply an $\ell_1$ penalty to perform implicit variable selection, while Ridge regression (Hoerl & Kennard, 1970) uses an $\ell_2$ penalty to control multicollinearity. Elastic Net (Zou & Hastie, 2005) combines both to balance sparsity and stability. Adaptations of these methods specifically for multi-response regression include imposing block-regularizations on $\mathbf{A}$ (Jalali et al., 2010), using Group Lasso to promote joint sparsity across responses (Yuan & Lin, 2006), and developing screening procedures for ultra-high-dimensional outputs (Kolar & Xing, 2010). Other methods exploit low-rank assumptions, such as the adaptive reduced-rank estimator of Wu et al. (2020), which prunes singular values to estimate effective model rank. Challenging the traditional reliance on convex relaxations, Bertsimas & Parys (2017) introduces exact cutting-plane algorithms to solve the $L_0$-constrained problem to optimality in high dimensions. Finally, Zhou & Zhao (2015) integrates task clustering to enforce shared sparsity across related responses.

**Representation learning methods** approach the problem by projecting predictors or responses into lower-dimensional latent spaces before regression. Classical approaches include Principal Component Regression (PCR) (Jolliffe, 1982) and Partial Least Squares (PLS) (Wold, 1982), which reduce dimensionality via linear transformations of the input features. More recent techniques, such as

Supervised Variational Autoencoders (S-VAE) (Kingma & Welling, 2014; Siddharth et al., 2017), learn task-aware nonlinear embeddings. Yet another approach is SHORE (Li et al., 2024), which compresses the high-dimensional output space $\mathbf{Y}$ under a sparsity assumption, enabling more efficient and stable estimation in extreme multi-output regimes.

**Clustering-based methods** capture structured dependencies by grouping predictors or responses. The core idea is that variables within a group share underlying characteristics, thus if predictors are clustered, each response can be predicted using only its relevant predictor cluster(s), reducing input dimensionality. Tree Lasso (Kim & Xing, 2010) models and clusters responses via overlapping group penalties, whereas Wang & Ye (2015) introduces screening rules tailored to tree-structured sparsity. The broader principle—leveraging group structure to reduce complexity—extends naturally to multi-task problems, as in Zhou & Zhao (2015), where similar tasks are clustered via so-called representative tasks.

While the aforementioned methods perform advanced high-dimensional regression, they rely on empirical statistical patterns or impose pre-specified structural constraints, such as fixed sparsity assumptions or pre-defined variable clusters, rather than explicitly modeling the latent causal mechanisms that might govern the relationships between variables. A crucial aspect often overlooked is the explicit modeling and exploitation of latent *causal* mechanisms that frequently underpin the observed relationships in complex, high-dimensional systems.

To this end, we introduce Causal Feature Augmentation (CFA), a novel framework for high-dimensional and low-sample multi-response regression rooted in a causality-informed perspective. In many real-world datasets, the dependencies between predictors $\mathbf{X}$ and responses $\mathbf{Y}$ are partly shaped by unobserved structure, such as latent mediators (where groups of predictors influence responses via an aggregated signal) or common confounders (where clusters of predictors and responses are jointly influenced by a shared hidden driver). Rather than attempting to recover the true causal graph, CFA uses these causal motifs as design principles to construct new, information-rich features that summarize shared $X$–$Y$ dependencies. These causality-inspired features are then integrated with the original predictors $\mathbf{X}$ to perform an augmented regression. Crucially, although this augmentation increases the total number of predictors fed into the model, it simplifies the underlying learning task. By explicitly representing these causally-derived features, the model can potentially achieve a far sparser and more robust solution, one that is less prone to overfitting and demonstrates superior generalization.

Our main contributions are threefold:

- We propose Causal Feature Augmentation (CFA), a new regression framework designed for high-dimensional and low-sample multivariate settings. CFA explicitly accounts for plausible latent causal structures by constructing mediator- and confounder-like features that summarize shared dependencies between predictors and responses, without requiring identifiability of the underlying latent variables.

- We develop a multi-stage algorithm for CFA that first extracts causally-related clusters of predictors based on their shared influence on responses. It then classifies these clusters into mediator or confounder structures, constructs new causal features accordingly, and solves an augmented regression problem using the causal-features. This approach results in models that are sparser, more robust, and achieve the same predictive performance in low-sample regimes that baselines require more samples to match.

- We provide empirical validation on synthetic datasets and multiple real-world datasets, including financial time series data. Our results demonstrate that CFA consistently and significantly outperforms a wide array of baselines.

## 2 CFA REGRESSION MODEL

In standard linear regression, we assume that the response variables $\mathbf{Y}$ are generated as a linear combination of the predictors $\mathbf{X}$, i.e., $\mathbf{Y} = \mathbf{X}\mathbf{A} + \mathbf{E}$, where $\mathbf{A}$ captures unknown coefficients and $\mathbf{E}$ denotes residual noise. The task is then to learn $\mathbf{A}$ from data using various regularization or estimation techniques as explained in the previous section. Implicit in this setup is a structural assumption about how data is generated—namely, that all dependencies between $\mathbf{X}$ and $\mathbf{Y}$ are direct and linear.

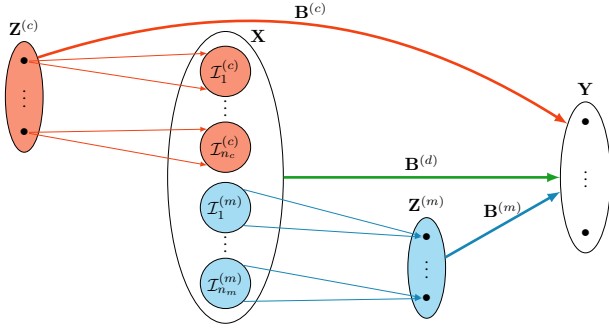
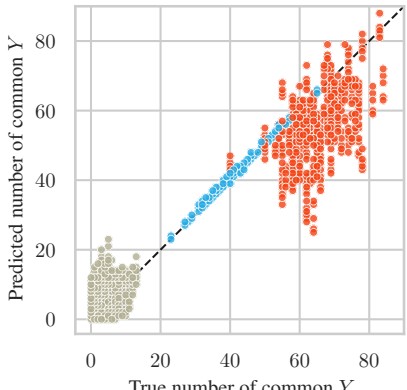

Figure 1: CFA model: confounders induce correlated clusters of predictors, mediators summarize subsets of predictors, and both affect the responses alongside direct effects from individual predictors.

Figure 2: Predicted vs true similarity scores for pairs of predictors in a synthetic dataset.

In this work, we adopt a more realistic, causally-informed perspective and propose a richer generative model that accounts for hidden causal pathways connecting $\mathbf{X}$ to $\mathbf{Y}$. Specifically, we posit that some predictors influence the responses through unobserved mediators or share latent common causes. This leads us to the following regression model:

$$\mathbf{Y} = \mathbf{X}\,\mathbf{B}^{(d)} \;+\; \mathbf{Z}^{(m)}\,\mathbf{B}^{(m)} \;+\; \mathbf{Z}^{(c)}\,\mathbf{B}^{(c)} \;+\; \mathbf{E}, \tag{1}$$

where $\mathbf{B}^{(d)} \in \mathbb{R}^{n_1 \times n_2}$ captures direct effects of $\mathbf{X}$ on $\mathbf{Y}$, $\mathbf{Z}^{(m)} \in \mathbb{R}^{N \times n_m}$ are *mediator* features (cluster-averages of some subsets of $\mathbf{X}$) with coefficients $\mathbf{B}^{(m)} \in \mathbb{R}^{n_m \times n_2}$, $\mathbf{Z}^{(c)} \in \mathbb{R}^{N \times n_c}$ are *confounder* features (latent variables causing some clusters of $\mathbf{X}$) with coefficients $\mathbf{B}^{(c)} \in \mathbb{R}^{n_c \times n_2}$, and $\mathbf{E}$ is the residual noise.

Figure 1 illustrates this generative structure, highlighting how predictors are organized into clusters either caused by latent confounders or summarized through mediators, both of which influence the response. This model enriches the regression setup by explicitly incorporating such causal structure, which we argue leads to simpler, sparser, and more generalizable regression solutions.

In the following subsections, we provide further details on these two causal structures—mediators and confounders—and how leveraging them improves prediction and interpretability.

## 2.1 MEDIATOR STRUCTURE

In many applications, responses often depend more on aggregated signals than on individual predictors. For instance in finance, equity returns are well known to co-move with sector or industry averages. Practitioners routinely use the mean return of all firms in a given sector as a key factor in forecasting individual stock performance.

Motivated by this, we assume that *some* of the predictors in $\mathbf{X}$ form $n_m$ disjoint *mediator clusters* $\mathcal{I}_1^{(m)}, \ldots, \mathcal{I}_{n_m}^{(m)} \subseteq \{1, \ldots, n_1\}$, not necessarily covering every variable. For each cluster $\mathcal{I}_i^{(m)}$, we define $Z_i^{(m)} = \frac{1}{|\mathcal{I}_i^{(m)}|} \sum_{j \in \mathcal{I}_i^{(m)}} X_j$, and $\mathbf{Z}^{(m)} = [\, Z_1^{(m)} \mid \cdots \mid Z_{n_m}^{(m)} \,]$. In the full model in eq. (1), the mediator coefficients $\mathbf{B}^{(m)}$ capture the influence of these averages on $\mathbf{Y}$, while $\mathbf{B}^{(d)}$ still allows for direct effects of all predictors (including those not in any mediator cluster). If the true relationship relies heavily on cluster-averages, many rows of $\mathbf{B}^{(d)}$ corresponding to clustered variables will become zero, and the total number of nonzeros in $(\mathbf{B}^{(d)}, \mathbf{B}^{(m)})$ can be much smaller than in a simple regression model that does not add mediators.

**Example 1.** *Suppose*

$$[Y_1 \mid Y_2 \mid Y_3] = [X_1 \mid X_2 \mid X_3 \mid X_4]\,\mathbf{A} \;+\; [e_1 \mid e_2 \mid e_3], \quad \mathbf{A} = \begin{pmatrix} \frac{1}{2} & \frac{1}{4} & 1 \\ \frac{1}{2} & \frac{1}{4} & 0 \\ \frac{1}{2} & 0 & \frac{1}{3} \\ \frac{1}{2} & 1 & \frac{1}{3} \end{pmatrix},$$

*with ten nonzeros. Define two mediator clusters $\mathcal{I}_1^{(m)} = \{1, 2\}$ and $\mathcal{I}_2^{(m)} = \{3, 4\}$, and*

$$Z_1^{(m)} = \frac{X_1 + X_2}{2}, \quad Z_2^{(m)} = \frac{X_3 + X_4}{2}, \quad \mathbf{Z}^{(m)} = [Z_1^{(m)} \mid Z_2^{(m)}].$$

*The augmented model eq. (1) becomes*

$$[Y_1 \mid Y_2 \mid Y_3] = [X_1 \mid X_2 \mid X_3 \mid X_4]\,\mathbf{B}^{(d)} \;+\; [Z_1^{(m)} \mid Z_2^{(m)}]\,\mathbf{B}^{(m)} \;+\; [e_1 \mid e_2 \mid e_3],$$

*with*

$$\mathbf{B}^{(d)} = \begin{pmatrix} 0 & 0 & 1 \\ 0 & 0 & 0 \\ 0 & 0 & 0 \\ 0 & 1 & 0 \end{pmatrix}, \quad \mathbf{B}^{(m)} = \begin{pmatrix} 1 & \frac{1}{2} & 0 \\ 1 & 0 & \frac{2}{3} \end{pmatrix}.$$

*By adding just these two mediator features, the total number of nonzero parameters drops from ten in $\mathbf{A}$ to six in $(\mathbf{B}^{(d)}, \mathbf{B}^{(m)})$, so a fit on the augmented model is potentially less prone to overfitting.*

In Section 3, we will present our procedure for discovering these unknown mediator clusters.

## 2.2 CONFOUNDER STRUCTURE

In many real-world settings, predictors in $\mathbf{X}$ exhibit strong correlations due to unobserved common causes—latent variables that simultaneously influence multiple features. These hidden variables, or *confounders*, often drive systematic dependencies among subsets of the predictors.

Crucially, such confounders frequently affect the responses $\mathbf{Y}$ as well. As a result, every predictor influenced by a shared confounder may appear marginally correlated with $\mathbf{Y}$, even if the true source of variation is the latent variable itself. This induces a situation where multiple predictors act as noisy proxies for the same underlying causal factor, which leads to unnecessarily denser direct-effect matrix $\mathbf{B}^{(d)}$ in eq. (1). If we were able to identify and explicitly recover these latent confounders, we could include them as features in the regression. Their effects would then be captured by $\mathbf{B}^{(c)}$, allowing $\mathbf{B}^{(d)}$ to focus on residual dependencies, yielding a sparser model.

Formally, we assume that some subset of predictors is partitioned into $n_c$ disjoint *confounder clusters* $\mathcal{J}_1^{(c)}, \ldots, \mathcal{J}_{n_c}^{(c)} \subseteq \{1, \ldots, n_1\}$, where each cluster $\mathcal{J}_i^{(c)}$ is influenced by an unobserved latent variable $Z_i^{(c)}$. We model the generative process for predictors in each cluster as

$$X_j = \gamma_j\, Z_i^{(c)} + \eta_j, \qquad \text{for } j \in \mathcal{J}_i^{(c)}, \tag{2}$$

where $\gamma_j$ is an unknown scalar coefficient indicating the strength of dependence of predictor $X_j$ on the latent confounder $Z_i^{(c)}$, and $\eta_j$ is the residual variation. We denote by $\mathbf{Z}^{(c)} \in \mathbb{R}^{N \times n_c}$ the matrix whose columns correspond to the confounder variables $Z_1^{(c)}, \ldots, Z_{n_c}^{(c)}$. When included in the regression model eq. (1), these features capture the shared variation via $\mathbf{B}^{(c)}$. We note that, unlike mediator features, which can be directly constructed as cluster averages, recovering latent confounders from data is more challenging. We will address this in Section 3.3.

## 3 METHOD

Given our regression model in the previous section, we now describe our algorithm for constructing the causally-informed features $\mathbf{Z}^{(m)}$ and $\mathbf{Z}^{(c)}$ and fitting the augmented regression model eq. (1). Our approach for solving the CFA model proceeds through four main steps:

1. **Causal clustering**: finds clusters over the predictor set $\{X_1, \ldots, X_{n_1}\}$ using a tailored similarity metric that captures latent mediator and confounder structure.[1]

2. **Cluster classification**: labels each cluster as a mediator cluster or a confounder cluster based on intra-cluster correlation patterns.

3. **Feature construction**: builds mediator features by averaging within mediator clusters, and recovers latent confounder features using PCA.

4. **Augmented regression**: solves the final regression in eq. (1) using Elastic Net.

We now describe each step in turn, noting that the core of our method is Step 1.

### 3.1 CAUSAL-CLUSTERING VIA NOVEL SIMILARITY METRIC

Our goal is to group predictors into clusters that correspond to either shared mediators or latent confounders. To this end, we define a pairwise similarity metric $s(i, j)$ between any two predictors $X_i, X_j$, which increases with the likelihood that they belong to the same cluster. More specifically, it directly quantifies their common associations with the response variables $Y_1, \ldots, Y_{n_2}$. We then apply a standard clustering algorithm (e.g., hierarchical clustering) to the similarity matrix $\mathbf{S}_{ij} = s(i, j)$.

#### 3.1.1 SIMILARITY METRIC DEFINITION

We define a pairwise similarity score between distinct predictors $X_i$ and $X_j$ as

$$s(i, j) = \big| \big\{ k : X_i \not\perp\!\!\!\perp Y_k \text{ and } X_j \not\perp\!\!\!\perp Y_k \big\} \big|, \tag{3}$$

i.e., the number of response variables that simultaneously depend on both $X_i$ and $X_j$. Thus $s(i, j)$ is an integer in $\{0, 1, \ldots, n_2\}$. In what follows, we show from a causal perspective that larger values of $s(i, j)$ are indicative of $X_i$ and $X_j$ belonging to the same mediator or confounder cluster. This makes $s(i, j)$ a suitable similarity metric for clustering, even though it does not by itself distinguish between mediators and confounders—this classification will be addressed in Section 3.2.

#### 3.1.2 THEORETICAL JUSTIFICATION OF THE SIMILARITY METRIC

Next, we provide a theoretical justification for the similarity metric $s(i, j)$ defined in eq. (3), and show that it is a meaningful proxy for clustering $X_i$ and $X_j$. Specifically, under reasonable structural assumptions, we will prove that

$$\mathbb{P}\big(X_i \not\perp\!\!\!\perp Y_k \text{ and } X_j \not\perp\!\!\!\perp Y_k \mid X_i \text{ and } X_j \text{ in the same cluster}\big)$$
$$> \mathbb{P}\big(X_i \not\perp\!\!\!\perp Y_k \text{ and } X_j \not\perp\!\!\!\perp Y_k \mid X_i \text{ and } X_j \text{ in different clusters}\big). \tag{4}$$

This inequality will justify the definition of $s(i, j)$: two predictors that belong to the same causal cluster (mediator or confounder) are more likely to exhibit joint dependence on the same response variables than predictors from different clusters.

To show this, we will first characterize all possible causal graph structures that can give rise to the joint dependencies $X_i \not\perp\!\!\!\perp Y_k$ and $X_j \not\perp\!\!\!\perp Y_k$. Then, we will analyze the likelihood of these structures given $X_i$ and $X_j$ being in the same cluster versus different clusters.

To this end, we enumerate in Table 1 all possible local dependency patterns that may arise under our generative model for a variable pair $(X_i, X_j)$, where both are dependent on a response $Y_k$. Note that we can group these structures using the five possible cluster assignments of the variable pair $(X_i, X_j)$: (i) whether $X_i$ and $X_j$ belong to the same or different clusters, and (ii) whether those clusters are of confounder type, mediator type, or mixed. For each of these five, a family of relevant causal subgraphs involving $(X_i, X_j, Y_k)$ is depicted in Table 1.

Now, for any fixed triple $(X_i, X_j, Y_k)$, let $\mathbb{G}_{ijk}$ denote the set of causal subgraphs of the true underlying model that match one of these patterns illustrated in Table 1. In the following theorem, proven in Appendix D, we formally state that these structures are indeed the only structures where both $X_i$ and $X_j$ are dependent on $Y_k$.

---

[1]While clusters are technically defined over predictor indices $\{1, \ldots, n_1\}$, we will often abuse notation and refer to clusters of predictors $\{X_1, \ldots, X_{n_1}\}$ for the sake of clarity.

**Theorem 3.1** (Dependence via causal graph structures). *Under the generative model described in Section 2, for any distinct pair of predictors $(X_i, X_j)$, and any response variable $Y_k$, we have:*

$$X_i \not\perp\!\!\!\perp Y_k \text{ and } X_j \not\perp\!\!\!\perp Y_k \quad \Longleftrightarrow \quad \mathbb{G}_{ijk} \text{ is non-empty.}$$

Next, we discuss the likelihood of these graphs under the following edge-generation process.

**Assumption 1.** *As a prior over cluster types, each predictor $X_j$ is independently assigned to a confounder cluster with probability $\pi_c$, and to a mediator cluster with probability $\pi_m = 1 - \pi_c$. Edges from predictors $X_j$ to responses $Y_k$ are independently included with probability $p_{xy}$; edges from mediators $Z_i^{(m)}$ to $Y_k$ with probability $p_{my}$; and edges from confounders $Z_i^{(c)}$ to $Y_k$ with probability $p_{cy}$. All edge-generation events are mutually independent. We assume that all edge probabilities are small (i.e., $p_{xy}, p_{my}, p_{cy} \ll 1$) and higher-order terms (e.g., $p_{xy}^2$) are negligible compared to lower-order terms such as $p_{my}$ and $p_{cy}$.*

This assumption is natural in high-dimensional settings: high-dimensional systems are well approximated by a few latent factors plus a sparse residual graph, so true direct $X \to Y$ edges are rare and events requiring two such edges occur with probability $\mathcal{O}(p_{xy}^2)$, negligible relative to first-order factor paths $(p_{my}, p_{cy})$ (Fan et al., 2013; Meinshausen & Bühlmann, 2006).

**Theorem 3.2** (Probabilistic justification of the similarity metric). *Under the edge-generation model of Assumption 1, Table 1 lists the probabilities of $\mathbb{G}_{ijk}$ being non-empty conditioned on each of the five cases of the table. These probabilities, which are a consequence of Assumption 1, imply*

$$\mathbb{P}\left(|\mathbb{G}_{ijk}| > 0 \mid X_i, X_j \text{ in the same cluster}\right) > \mathbb{P}\left(|\mathbb{G}_{ijk}| > 0 \mid X_i, X_j \text{ in different clusters}\right),$$

*where $|\mathbb{G}_{ijk}|$ denotes the cardinality of the set of causal subgraphs compatible with the underlying model of Section 2.*

Theorems 3.1 and 3.2 together imply eq. (4), which justifies $s(i, j)$ as an apt similarity metric for clustering predictors.

To further validate our theoretical findings, Figure 2 visualizes the similarity scores for a synthetic dataset generated according to our model in Section 2, with the same configuration we use for the synthetic experiments in Section 4. Each dot corresponds to a pair $(X_i, X_j)$, colored orange if both belong to the same confounder cluster, blue if both are in the same mediator cluster, and gray otherwise. The $x$-axis shows the true similarity score (computed from the ground-truth graph), while the $y$-axis shows the empirical similarity (estimated from samples). While sampling noise and imperfect statistical tests introduce deviations from the diagonal $y = x$, the separation is clear: orange and blue points tend to lie to the right of gray ones, confirming our theoretical prediction that same-cluster pairs tend to share more response dependencies.

### 3.2 CLUSTER CLASSIFICATION

To determine whether a given predictor cluster corresponds to a mediator or a confounder, we leverage intra-cluster correlation patterns. In a confounder cluster, the shared latent variable induces correlation among all predictors, whereas in a mediator cluster, no such correlation is expected among its members. Therefore, in our method, for each detected cluster, we compute all pairwise correlations between its member variables and perform a two-tailed hypothesis test, where the null hypothesis is that each correlation is zero. If the null hypothesis is rejected for a significant proportion of pairs, we classify the cluster as a confounder; otherwise, we classify it as a mediator.

### 3.3 FEATURE CONSTRUCTION

Given the clustered and classified predictors, we now construct the augmented features $\mathbf{Z}^{(m)}$ and $\mathbf{Z}^{(c)}$ used in the regression model eq. (1). For each mediator cluster $\mathcal{I}_i^{(m)}$, we define the corresponding feature as a simple average: $Z_i^{(m)} = \frac{1}{|\mathcal{I}_i^{(m)}|} \sum_{j \in \mathcal{I}_i^{(m)}} X_j$.

Confounder features are recovered differently. Recall the generative model eq. (2), where each $X_j$ in a confounder cluster is a noisy linear function of a latent variable $Z_i^{(c)}$ with unknown loadings. For each

confounder cluster $\mathcal{I}_i^{(c)}$, we estimate $Z_i^{(c)}$ as the first principal component of the subset of predictors in that cluster. This procedure is well motivated under a linear factor model with independent noise: when the confounder induces the dominant direction of variation, PCA consistently estimates the latent variable up to scale (Anderson, 1958; Tipping & Bishop, 1999; Paul, 2007). Importantly, the true confounder is not identifiable in absolute terms—only its direction up to scaling can be recovered. However, this is sufficient for our purposes, since the scale ambiguity is absorbed into the learned coefficients $\mathbf{B}^{(c)}$ during regression.

### 3.4 Augmented regression

Having constructed the mediator and confounder features $\mathbf{Z}^{(m)}$ and $\mathbf{Z}^{(c)}$, we now augment the original predictors $\mathbf{X}$ with these new variables and solve a regularized regression problem. In our implementation, we use Elastic Net (Zou & Hastie, 2005), which combines both $\ell_1$ and $\ell_2$ regularization to promote sparsity while mitigating instability in correlated settings. Concretely, given $\mathbf{X}, \mathbf{Z}^{(m)}, \mathbf{Z}^{(c)}$, we solve:

$$
\begin{aligned}
\min_{\mathbf{B}^{(d)},\mathbf{B}^{(m)},\mathbf{B}^{(c)}} \quad & \frac{1}{2N}\left\|\mathbf{Y} - \mathbf{X}\mathbf{B}^{(d)} - \mathbf{Z}^{(m)}\mathbf{B}^{(m)} - \mathbf{Z}^{(c)}\mathbf{B}^{(c)}\right\|_F^2 \\
& + \lambda_d^{(1)}\|\mathbf{B}^{(d)}\|_{1,1} + \lambda_m^{(1)}\|\mathbf{B}^{(m)}\|_{1,1} + \lambda_c^{(1)}\|\mathbf{B}^{(c)}\|_{1,1} \\
& + \lambda_d^{(2)}\|\mathbf{B}^{(d)}\|_F^2 + \lambda_m^{(2)}\|\mathbf{B}^{(m)}\|_F^2 + \lambda_c^{(2)}\|\mathbf{B}^{(c)}\|_F^2.
\end{aligned}
\tag{5}
$$

Note that our core contribution is the augmentation with causally motivated features; the downstream regressor can be any suitable linear model—Elastic Net is a natural choice.

### 3.5 Scope of CFA

It is important to emphasize that CFA does not attempt causal discovery, nor does it require (or claim) identifiability or recovery of the true mediators, confounders, or causal graph. Instead, we use simple causal generative models as a design tool for constructing features that can improve prediction and robustness in high-dimensional, low-sample regimes. If the augmented features $\mathbf{Z}^{(m)}, \mathbf{Z}^{(c)}$ are uninformative, the Elastic Net penalty shrinks their coefficients toward zero, so performance matches a strong regularized baseline. When approximate mediator/confounder patterns are present, the augmentation captures shared variation and reduces effective complexity, yielding sparser solutions and better generalization—even if the augmented features do not coincide with any unobservable ground truth. In abundant-data regimes, the gap between CFA and Elastic Net narrows, with the most significant gains from CFA arising in low-sample settings, precisely our target regime.

It is worth noting that while our derivation assumes ancestral predictors, the CFA framework generalizes to anti-causal settings where responses cause predictors via latent mechanisms (e.g., $Y \rightarrow Z^{(am)} \rightarrow \mathbf{X}$). In such cases, $Z^{(am)}$ induces correlation among $\mathbf{X}$, leading CFA to classify the group as a 'confounder' cluster. PCA then recovers $Z^{(am)}$ as a predictive feature for $Y$. Thus, CFA effectively captures both causal and anti-causal latent structures.

## 4 Experiments

In this section, we present experiments on a synthetic dataset and real-world datasets to evaluate the performance of CFA against several baselines. We provide further details on the setup and implementation in Appendix A. For the sake of space, we include additional experiments on four real-world datasets and true graph recovery in Appendix B, and time complexity analysis in Appendix C.

### 4.1 Baselines

We compare our CFA method against several baselines, categorized as: (i) optimization-based, including `ElasticNet` Zou & Hastie (2005), `Lasso` Tibshirani (1996), and `Ridge` Hoerl & Kennard (1970); (ii) clustering-based, represented by a custom Cluster Regressor (`CR`) that groups predictors based on their correlation and trains separate ElasticNet models for each response using only

Table 1: Causal subgraphs and their likelihood probabilities for triplet $(X_i, X_j, Y_k)$, where $X_i$ and $X_j$ are pairwise dependent with $Y_k$, categorized by cluster assignment of $X_i$ and $X_j$. Solid edges mark the connections that must exist, while missing edges remain permissible. Dashed edges indicate that at least one of the variables in the dashed box must connect to $Y_k$.

| | **Same cluster** ($t_1 = t_2$) | **Different cluster** ($t_1 \neq t_2$) |
|---|---|---|
| $i \in \mathcal{I}_{t_1}^{(m)}, j \in \mathcal{I}_{t_2}^{(m)}$ | $p_{my} + (1-p_{my})p_{xy}^2 \approx p_{my}$ | $\left(1 - (1-p_{xy})(1-p_{my})\right)^2 \approx (p_{my} + p_{xy})^2$ |
| $i \in \mathcal{J}_{t_1}^{(c)}, j \in \mathcal{J}_{t_2}^{(c)}$ | $1 - (1-p_{cy})(1-p_{xy})^{m_c} \approx p_{cy} + m_c p_{xy}$ | $\left(1 - (1-p_{cy})(1-p_{xy})^{m_c}\right)^2 \approx (p_{cy} + m_c p_{xy})^2$ |
| $i \in \mathcal{I}_{t_1}^{(m)}, j \in \mathcal{J}_{t_2}^{(c)}$ | Impossible | $\left(1 - (1-p_{xy})(1-p_{my})\right)\left(1 - (1-p_{cy})(1-p_{xy})^{m_c}\right)$ $\approx (p_{my} + p_{xy})(p_{cy} + m_c p_{xy})$ |

predictors from its corresponding cluster; and (iii) representation learning-based, including `SHORE` Li et al. (2024), which first compresses the high-dimensional output space into a latent representation, performs regression, and then reconstructs sparse outputs, Principal Component Regression (`PCR`) Jolliffe (1982), Partial Least Squares Regression (`PLS`) Wold (1982), and Supervised Variational Autoencoder (`S-VAE`) Kingma & Welling (2014). Full details are given in Appendix A.1.

## 4.2 Synthetic dataset

We consider a synthetic multi-task regression dataset generated from the model in Section 2. More specifically, we consider a high-dimensional setting with $n_1 = n_2 = 1000$ predictors and responses, $n_m = 250$ mediator clusters, and $n_c = 250$ confounder clusters. We randomly sample predictors, confounders, and noise from a normal distribution with varying variances. For complete details on the data generation process, please refer to Appendix A.2.

We use an 80%/20% split for training, and test sets, respectively, and use 3-fold cross-validation on the training set to tune hyperparameters. We use the correlation between the predicted and true responses as the metric. Full details on the experimental setup are provided in Appendix A.1.

**Results.** Figure 3 illustrates the test correlation as a function of the number of samples $N \in [100, 5000]$ on our synthetic dataset. CFA (blue solid line) consistently outperforms all baselines across the entire range of sample sizes evaluated. While several optimization-based methods (Ridge, Lasso, ElasticNet) and the representation-learning-based PCR also eventually converge to a perfect correlation as $N$ increases, CFA gets there much more rapidly, particularly in the critical low-sample regime, which is precisely the target regime of CFA.

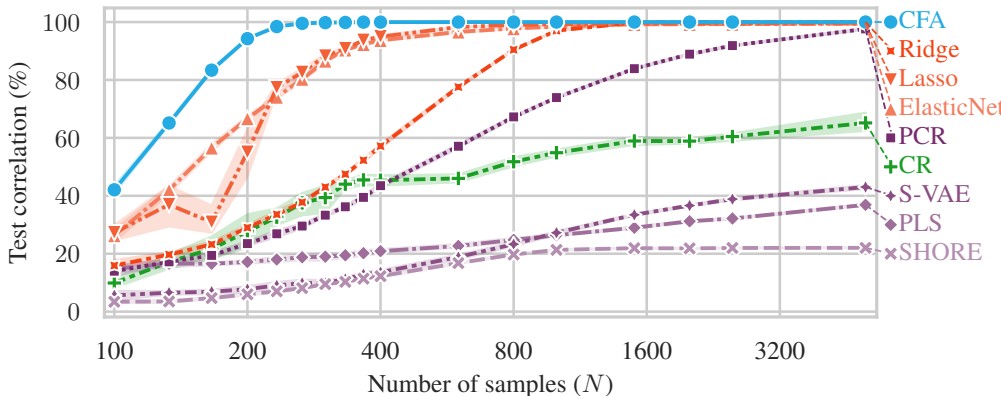

Figure 3: Test correlation of each algorithm versus changing sample size $N$ (in log-scale) on the synthetic dataset. The shaded area represents the 95% confidence interval.

For instance, with merely $N = 160$ total samples (corresponding to 112 training samples), CFA attains a test correlation of 83%. This is substantially (50%) higher than the next best performing baseline, ElasticNet, which achieves 56% under the same conditions. Other approaches, including the clustering-based CR and further representation learning methods like S-VAE, PLS, and SHORE, exhibit significantly lower performance, especially at smaller sample sizes. This demonstrates CFA's superior sample efficiency and robustness against overfitting, particularly when data is scarce.

### 4.3 REAL-WORLD DATASET: S&P 500 STOCK RETURNS

Real-world data often exhibits complex dependencies among predictors and responses, ones that may not even be modeled by our assumed generative process. However, our central hypothesis is that even in messy, real-world systems, patterns that approximate our mediator and confounder structures are common and highly predictive. CFA then acts as a powerful pattern-matcher, extracting these "good enough" proxies to produce more robust and accurate predictions. To evaluate this hypothesis, we first apply our method to a financial dataset, and also on four other real-world datasets in Appendix B.1.

We evaluate our method on data derived from daily closing prices of S&P 500 constituents for the period of 2020–2025, with the goal of predicting next-day log returns of each stock using yesterday's log returns of all stocks. Predicting these returns is notoriously difficult because of high market efficiency, inherent noise, and complex interdependencies. To capture varying market conditions, we create multiple datasets using a rolling three-year window. For each window, we select stocks with complete price data, ranging from 479 to 498, forming a multi-task regression problem where $n_1 = n_2$ equals the number of selected stocks for that period. The number of samples $N$ is the number of trading days in the window, typically around 750, except the last window, which has 680 samples. The dataset is split chronologically into training, validation, and test sets, with a split of 70%/15%/15%. A detailed description of the data preprocessing steps is provided in Appendix A.3.

**Results.** Table 2 summarizes the train and test correlations between predicted and target log returns. CFA consistently delivers strong out-of-sample performance, achieving the highest test correlation in all four windows and showing smaller train–test gaps and thus reduced overfitting than strong baselines; for example, Ridge often records higher training scores but does not generalize as well. SHORE underperforms, likely due to its inherent assumption of sparse outputs, a characteristic typically absent in financial time-series data.

Note that while the test correlations appear low, predicting next-day S&P 500 returns is notoriously difficult due to market efficiency and high noise. Thus, in this literature, low single-digit out-of-sample correlations are already meaningful and the 2–3% test correlations in Table 2 reflect genuine signal rather than noise, making CFA's consistent edge practically relevant.

Table 2: Train and test Pearson correlations (%) across rolling three-year windows on the S&P 500 dataset. Each cell reports the correlation $\pm$ uncertainty, where uncertainty is the Fisher-$z$ 95% CI width. Bold indicates the highest test within each window. Rows sorted by test correlation.

| Window | 2020-2022 | | 2021-2023 | | 2022-2024 | | 2023-2025 | |
|---|---|---|---|---|---|---|---|---|
| Metric | Train | Test | Train | Test | Train | Test | Train | Test |
| **CFA** | $77.27_{\pm 1.62}$ | $\mathbf{1.76}_{\pm 0.13}$ | $51.60_{\pm 2.91}$ | $\mathbf{1.38}_{\pm 0.12}$ | $68.54_{\pm 2.12}$ | $\mathbf{3.17}_{\pm 0.09}$ | $81.75_{\pm 1.34}$ | $\mathbf{2.92}_{\pm 0.09}$ |
| **ElasticNet** | $66.63_{\pm 2.22}$ | $1.51_{\pm 0.19}$ | $62.37_{\pm 2.44}$ | $0.87_{\pm 0.21}$ | $62.36_{\pm 2.44}$ | $2.32_{\pm 0.22}$ | $73.73_{\pm 1.83}$ | $2.37_{\pm 0.17}$ |
| **Ridge** | $94.73_{\pm 0.42}$ | $1.41_{\pm 0.21}$ | $92.07_{\pm 0.62}$ | $1.03_{\pm 0.18}$ | $92.07_{\pm 0.62}$ | $1.84_{\pm 0.27}$ | $97.94_{\pm 0.17}$ | $2.19_{\pm 0.20}$ |
| **CR** | $22.20_{\pm 3.74}$ | $1.08_{\pm 0.27}$ | $21.03_{\pm 3.76}$ | $1.36_{\pm 0.12}$ | $17.72_{\pm 3.81}$ | $0.68_{\pm 0.36}$ | $40.01_{\pm 3.32}$ | $2.38_{\pm 0.17}$ |
| **PCR** | $58.73_{\pm 2.61}$ | $1.00_{\pm 0.28}$ | $1.25_{\pm 3.93}$ | $-0.27_{\pm 0.35}$ | $72.98_{\pm 1.87}$ | $1.80_{\pm 0.28}$ | $75.13_{\pm 1.75}$ | $2.07_{\pm 0.21}$ |
| **PLS** | $50.29_{\pm 2.96}$ | $1.17_{\pm 0.25}$ | $51.45_{\pm 2.92}$ | $-1.36_{\pm 0.39}$ | $48.11_{\pm 3.05}$ | $1.38_{\pm 0.32}$ | $43.26_{\pm 3.22}$ | $0.00_{\pm 0.38}$ |
| **S-VAE** | $54.20_{\pm 2.81}$ | $0.01_{\pm 0.38}$ | $0.11_{\pm 3.93}$ | $0.22_{\pm 0.30}$ | $1.98_{\pm 3.93}$ | $-0.26_{\pm 0.39}$ | $39.94_{\pm 3.32}$ | $0.99_{\pm 0.32}$ |
| **SHORE** | $37.26_{\pm 3.40}$ | $0.55_{\pm 0.34}$ | $32.15_{\pm 3.54}$ | $0.00_{\pm 0.33}$ | $25.16_{\pm 3.69}$ | $0.43_{\pm 0.37}$ | $30.75_{\pm 3.57}$ | $1.21_{\pm 0.30}$ |
| **Lasso** | $20.60_{\pm 3.77}$ | $-0.52_{\pm 0.39}$ | $8.44_{\pm 3.90}$ | $-0.03_{\pm 0.33}$ | $8.03_{\pm 3.90}$ | $-0.48_{\pm 0.39}$ | $18.45_{\pm 3.80}$ | $-0.86_{\pm 0.39}$ |

## 4.4 ADDITIONAL REAL-WORLD DATASETS

In addition to the S&P 500 experiments, we further evaluate CFA on four additional real-world multi-output regression datasets from the Mulan repository (Spyromitros-Xioufis et al., 2016):

- ATP-7D (airline ticket prices), where the task is to predict minimum ticket prices for multiple flight preferences over the next 7 days.
- OES-10 (occupational employment survey), which predicts employment counts for multiple job types across US cities.
- RF-2 (river flow), which forecasts river flows at multiple sites 48 hours ahead.
- SCM-1D (supply chain management), which predicts next-day prices for multiple products in a simulated trading environment. These datasets cover heterogeneous domains (transportation, labor economics, hydrology, and markets) and mixed feature types.

We use the same evaluation metric (average Pearson correlation across outputs) and the same set of baselines as in the main experiments. Full train and test correlations for all methods are reported in Table 4 in Appendix B.1. Here we summarize the main findings: across all four datasets, CFA attains the highest test correlation, improving over the strongest baseline by roughly 3–5 absolute correlation points. At the same time, CFA avoids the severe overfitting observed for methods such as Ridge and Lasso, which often reach near-perfect training scores but substantially lower test performance. These results indicate that the gains we observe on the synthetic and S&P 500 experiments transfer to diverse real-world settings and are not specific to a single application domain.

## 5 CONCLUSION

We introduced Causal Feature Augmentation, a regression framework designed for high-dimensional, multi-output regression by explicitly modeling latent causal structures—namely unobserved mediators and confounders. We then proposed a multi-stage algorithm, CFA, that extracts these causal relationships, constructs corresponding features, and solves an augmented regression problem. Experiments on both synthetic and real-world datasets demonstrate that CFA outperforms baselines by yielding sparser, more robust models with superior generalization. Promising future directions include extending CFA to non-linear settings, incorporating soft clustering to capture overlapping causal structure, and exploring alternative augmented regression formulations.

## REPRODUCIBILITY STATEMENT

We have taken several steps to ensure reproducibility. An anonymous repository with Python source code is provided at `https://anonymous.4open.science/r/CFA`. The source code includes implementations of our CFA method in addition to all tested baselines, data generation scripts, and experiment scripts. Moreover, the code is well-documented using the Google style. The full experimental setup, including hardware specifications, dataset details, hyperparameter tuning,

and algorithmic specifics, is described in Appendix A. All theoretical assumptions and complete proofs are provided in Appendix D.

## LLM USAGE

We used LLMs as general-purpose assistive tools for (i) language polishing and paraphrasing, (ii) drafting and refining `TikZ` figures from our own sketches/specifications, and (iii) writing and debugging non-core code (e.g., experiment scripts, plotting utilities, small refactors). LLMs did *not* contribute to research ideation, problem formulation, algorithmic design, theoretical results, or the interpretation of experiments. Thus, LLM usage here was not significant enough to warrant authorship.

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

# Appendix

## A   ADDITIONAL EXPERIMENTAL DETAILS

Here, we provide more details on the experiments presented in Section 4. Note that all experiments, coded in Python 3.11, were conducted on a machine equipped two Intel Xeon E5-2680 v3 CPUs, 256GB of RAM, and running Ubuntu 24.04.1 LTS.

**Metric**   We evaluate the predictive performance of all models using the average Pearson correlation coefficient. Given the true target matrix $\mathbf{Y} \in \mathbb{R}^{N \times n_2}$ and the corresponding prediction matrix $\widehat{\mathbf{Y}} \in \mathbb{R}^{N \times n_2}$, where $N$ is the number of samples and $n_2$ is the number of response variables (or tasks), we first compute the Pearson correlation for each individual response variable. Specifically, for each column $j$ (where $j = 1, \ldots, n_2$), we calculate $\rho_j = \text{corr}(\mathbf{Y}_{:,j}, \widehat{\mathbf{Y}}_{:,j})$, where $\mathbf{Y}_{:,j}$ and $\widehat{\mathbf{Y}}_{:,j}$ are the $j$-th columns of the true and predicted matrices, respectively. The final reported metric is the arithmetic mean of these individual correlation coefficients. This metric ranges from -1 to 1, where 1 indicates perfect positive linear correlation, -1 indicates perfect negative linear correlation, and 0 indicates no linear correlation. Higher values signify better predictive accuracy.

### A.1   ALGORITHMS

Here, we provide the details of the parameters used and tuned for our approach as well as each baseline. For all methods, hyperparameters are tuned using Hyperopt Bergstra et al. (2013),[2] with the Tree-structured Parzen Estimator (TPE), minimizing negative Pearson correlation on the validation set for the S&P 500 dataset, and 3-fold cross validation for others. The number of Hyperopt evaluations is 50.

- **CFA**: For dependency testing between predictors and responses, we use a thresholded 2-tailed correlation $t$-test. We tune its $p$-value threshold in $[0.01, 0.1]$. We use bottom-up hierarchical clustering with average linkage, for which we tune the number of clusters $n_c + n_m \in [2, n_1/5]$. For cluster classification, we also tune the $p$-value threshold for the independence test in $[0.01, 0.1]$. After clustering, we use the Elastic Net regressor, detailed below, to solve the augmented regression problem. To ensure a fair comparison with Elastic Net, we constrained the regularization to be uniform across all feature types, using a single, global L1/L2 ratio and a single overall regularization strength for the combined feature set $[\mathbf{X}, \mathbf{Z}^{(m)}, \mathbf{Z}^{(c)}]$. As a result, the final regression step in our implementation of CFA has the exact same number of hyperparameters to tune as the standard Elastic Net baseline.
- **ElasticNet**: We use the Elastic Net regressor from Scikit-learn Pedregosa et al. (2011),[3] which combines $L_1$ and $L_2$ penalties. We tune the regularization strength $\alpha \in [10^{-6}, 10^2]$ and $\ell_1$-ratio $\in [0, 1]$.
- **Lasso**: We use the Lasso regressor Tibshirani (1996) from Scikit-learn, which uses only the $L_1$ penalty. We tune the regularization strength $\alpha \in [10^{-6}, 10^2]$.
- **Ridge**: We use the Ridge regressor Hoerl & Kennard (1970) from Scikit-learn, which uses only the $L_2$ penalty. We tune the regularization strength $\alpha \in [10^{-6}, 10^2]$.
- **SHORE**: The SHORE (Sparse & High-dimensional-Output REgression) regressor Li et al. (2024) addresses multi-output regression by first compressing the high-dimensional output space into a lower-dimensional latent space. It then learns a linear regressor in this compressed space and finally predicts sparse, high-dimensional outputs through a specific iterative algorithm. We use their official implementation.[4] We tune its prediction singular value threshold (integer in $[10, 100]$), number of compressed dimensions (integer in $[10, n_1/5]$), and PGD learning rate ($[10^{-4}, 10^{-1}]$).
- **PCR (Principal Component Regression)**: PCA is first applied to the predictors $\mathbf{X}$, followed by an ElasticNet regressor on the principal components. We tune the number of components in PCA to retain a proportion of variance in $[0.8, 0.99]$, and the Elastic Net's hyperparameters as described above.

---

[2]https://hyperopt.github.io/hyperopt/

[3]https://scikit-learn.org

[4]https://github.com/renyuanli/Solving_SHORE_via_compression/tree/main

- **CR (ClusterRegressor)**: A custom baseline for time-series forecasting where predictors $\mathbf{X}$ are first clustered based on their Pearson correlation matrix. For each response $Y_j$, an ElasticNet model is trained using only the predictors $X_i$ belonging to the same cluster as $X_j$. We tune the number of clusters, clustering method (hierarchical or spectral), and for each ElasticNet, its regularization strength $\alpha \in [10^{-1}, 10^1]$ (log-uniform) and $\ell_1$-ratio $\in [0, 1]$ (uniform).

- **PLS (Partial Least Squares Regression)**: We use the PLS regression implementation of Scikit-learn and we tune the number of PLS components (integer in $[2, n_1/5]$ for synthetic experiments).

- **S-VAE (Supervised Variational Autoencoder)**: A VAE model with an additional regression head that predicts $\mathbf{Y}$ from the latent space $Z$, trained end-to-end. We tune the encoder/decoder hidden layer dimensions (choices from pre-defined architectures like $[128, 64]$), latent space dimensionality (integer in $[5, n_1/5]$), learning rate ($[10^{-6}, 10^{-1}]$), and the weights for VAE loss $\alpha_{VAE} \in [10^{-3}, 10^1]$ and supervised loss $\beta_{sup} \in [10^{-3}, 10^1]$.

## A.2 SYNTHETIC DATASET

The synthetic dataset is generated using the following procedure:

1. **Latent Confounders ($\mathbf{Z}^{(c)}$)**: $n_c$ latent confounder variables $Z_k^{(c)}$ are drawn independently, with each $Z_k^{(c)} \sim \mathcal{N}(0, \sigma_{z_c,k}^2)$ where standard deviations $\sigma_{z_c,k}$ are sampled uniformly from $U(2, 4)$.

2. **Predictors ($\mathbf{X}$)**: The $n_1$ predictors are randomly assigned to one of the $n_m + n_c$ clusters.
   - For $X_i$ in a confounder cluster $k$: $X_i = w_i Z_k^{(c)} + \epsilon_{X_i}$, where the weight $w_i \sim U(2, 4)$ and $\epsilon_{X_i} \sim \mathcal{N}(0, \sigma_{x,i}^2)$ with $\sigma_{x,i} \sim U(2, 4)$.
   - For $X_i$ in a mediator cluster: $X_i \sim \mathcal{N}(0, \sigma_{x_m,i}^2)$ with $\sigma_{x_m,i} \sim U(5, 7)$ to ensure these predictors have substantial variance.

3. **Mediator Features ($\mathbf{Z}^{(m)}$)**: The $n_m$ mediator features $Z_k^{(m)}$ are constructed by averaging the predictors $X_i$ within each respective mediator cluster.

4. **Responses ($\mathbf{Y}$)**: The $n_2$ responses are generated via the linear model $\mathbf{Y} = \mathbf{X}\mathbf{B}^{(d)} + \mathbf{Z}^{(m)}\mathbf{B}^{(m)} + \mathbf{Z}^{(c)}\mathbf{B}^{(c)} + \mathbf{E}$. The coefficient matrices $\mathbf{B}^{(d)} \in \mathbb{R}^{n_1 \times n_2}$, $\mathbf{B}^{(m)} \in \mathbb{R}^{n_m \times n_2}$, and $\mathbf{B}^{(c)} \in \mathbb{R}^{n_c \times n_2}$ are sparse. Non-zero entries are drawn from $U([-1.5, -0.5] \cup [0.5, 1.5])$ for $\mathbf{B}^{(d)}$ and $\mathbf{B}^{(c)}$, and from $U([-3, -1] \cup [1, 3])$ for $\mathbf{B}^{(m)}$ (reflecting stronger mediator effects). The sparsity pattern is determined by probabilities $p_{xy} = 0.006$ for $\mathbf{X} \to \mathbf{Y}$ direct effects, $p_{z^m y} = 0.04$ for $\mathbf{Z}^{(m)} \to \mathbf{Y}$ mediator effects, and $p_{z^c y} = 0.04$ for $\mathbf{Z}^{(c)} \to \mathbf{Y}$ confounder effects. The final noise $\mathbf{E}$ consists of columns $\mathbf{E}_j \sim \mathcal{N}(0, \sigma_{\text{noise},j}^2)$, where $\sigma_{\text{noise},j} \sim U(1, 2)$.

## A.3 S&P 500 DATASET

We generate a series of datasets using a rolling window approach, starting from 2020-01-01 until 2025-08-30. Each dataset corresponds to a three-year period of market data (e.g., 2020-01-01 to 2022-12-31 for the "2020-2022" window), with the start year of these windows rolling forward annually from 2020 up to 2023. For each three-year window, we perform the following processing steps:

1. **Stock Selection**: We retain only those S&P 500 stocks that have complete daily closing price data throughout the specific three-year window. This results in a varying number of stocks ($n_s$) for each dataset window, where $n_1 = n_2 = n_s$. The average number of stocks across all datasets is approximately 495.

2. **Predictor Generation ($\mathbf{X}$)**: The predictors are based on past 1-day log returns. For each stock $i$ and day $t$, the daily log return is $r_{i,t} = \log(P_{i,t}/P_{i,t-1})$. These returns are then cross-sectionally demeaned for each day $t$ (i.e., $\tilde{r}_{i,t} = r_{i,t} - \frac{1}{n_s}\sum_{j=1}^{n_s} r_{j,t}$). Finally, an Exponentially Weighted Moving Average (EWMA) with a halflife of 3 days is applied to these demeaned log returns to form the predictor features $X_{i,t}$.

3. **Response Generation ($\mathbf{Y}$)**: The responses are the 1-day ahead future log returns. For each stock $i$ and day $t$, the future log return is $y_{i,t} = \log(P_{i,t+1}/P_{i,t})$. These future returns are also cross-sectionally demeaned, similar to the predictors.

Table 3: The number of stocks and trading days in each three-year window of the S&P 500 dataset.

| Window | 2020–2022 | 2021–2023 | 2022–2024 | 2023–2025 |
|---|---|---|---|---|
| **Nr. of stocks** ($n_1 = n_2$) | 490 | 495 | 496 | 498 |
| **Nr. of trading days** ($N$) | 754 | 751 | 750 | 680 |

4. **Alignment and Cleaning**: Days for which either predictor or response data are incomplete (due to initial shifts or EWM calculations) are removed.

5. **Train-Validation-Test Split**: For each three-year window dataset, the available daily samples are split chronologically into training, validation, and testing sets. The first 70% of the temporal data forms the training set, the subsequent 15% and 15% constitute the validation and test sets, ensuring no future information leakage.

6. **Standardization**: Both the training predictors $\mathbf{X}_{\text{train}}$ and training responses $\mathbf{Y}_{\text{train}}$ are standardized independently (to have zero mean and unit variance). The scaling parameters learned from the training set are then applied to standardize $\mathbf{X}_{\text{test}}$ and $\mathbf{Y}_{\text{test}}$.

Table 3 summarizes the number of stocks and trading days in each three-year window of the S&P 500 dataset. The number of stocks varies slightly across windows due to the rolling nature of the dataset, while the number of trading days is consistent across most windows, with a slight reduction in the last window due to the incomplete data at the time of writing.

## B   ADDITIONAL EXPERIMENTAL RESULTS

### B.1   MIXED REAL-WORLD DATASETS

We further evaluate our method on four other real-world datasets (Spyromitros-Xioufis et al., 2016) from diverse domains, with mixtures of continuous, discrete, and categorical data. These are:

- **Airline ticket prices (ATP-7D):** This time-ordered dataset involves predicting the minimum ticket price over the next 7 days for 6 different flight preferences. The 411 input features are highly heterogeneous, including the number of days until departure, boolean day-of-the-week indicators, and a wide array of price and quote statistics from multiple airlines and stopover options, providing a rich mix of variable types.

- **Occupational employment survey (OES-10):** Sourced from the US Bureau of Labor Statistics, this dataset presents a cross-sectional regression task. The goal is to predict the number of full-time employees for a set of target job types within 403 different US cities. The input variables are the employment numbers for other job categories, creating a high-dimensional problem where latent economic factors likely confound employment across different sectors.

- **River flow (RF-2):** This dataset contains hourly time-series data from 8 sites in the Mississippi River network. The task is to predict river flows 48 hours into the future, using past flow observations from all sites as well as discrete precipitation forecasts. The physical connectivity of the river network provides a strong real-world basis for the existence of mediator and confounder structures.

- **Supply chain management (SCM-1D):** Derived from a complex trading agent competition, this dataset requires predicting the next-day mean price for 16 different products. The features include current and time-lagged prices from a simulated economy with 18 competing games.

We use a 80%/20% split for training and testing. We tune the hyperparameters of each method using TPE and 3-fold cross-validation on the training set, selecting the best configuration based on validation performance. We then retrain the model on the full training set with these hyperparameters and evaluate on the test set.

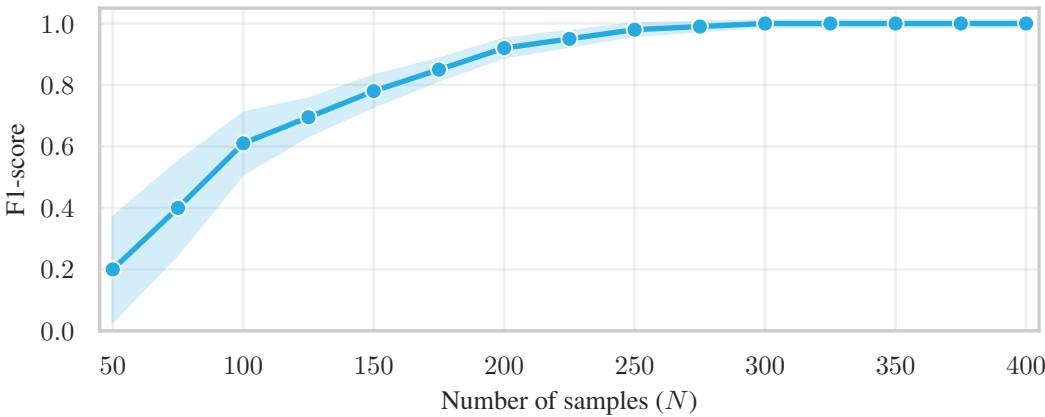

Figure 4: F1-score of CFA in recovering the true latent graph on synthetic data, varying the number of samples $N$. The shaded area represents the 95% confidence interval over 20 runs.

**Results.** The final training and test Pearson correlations are reported in Table 4. Across all four datasets, CFA delivers the highest test-set performance. Crucially, its training performance is not always the highest; methods like Ridge and Lasso achieve near-perfect training scores but generalize poorly, indicating severe overfitting. CFA's ability to find a solution that is both highly accurate and robust demonstrates its superior ability to prevent overfitting. This supports our central hypothesis: simply regularizing the learning process is less effective than proactively identifying latent structures and constructing new, information-rich causal features. By simplifying the underlying learning task, CFA enables the model to achieve better and more reliable generalization.

Table 4: Train and test Pearson correlations (%) on datasets from Mulan (Spyromitros-Xioufis et al., 2016). Each cell reports value $\pm$ uncertainty, where uncertainty is the Fisher-$z$ 95% CI width. Bold indicates the highest test within each window. Rows sorted by test correlation.

| | ATP-7D | | OES-10 | | RF-2 | | SCM-1D | |
|---|---|---|---|---|---|---|---|---|
| | Train | Test | Train | Test | Train | Test | Train | Test |
| **CFA** | $93.5 \pm 1.6$ | $\mathbf{80.3} \pm 4.4$ | $98.0 \pm 0.5$ | $\mathbf{96.1} \pm 1.0$ | $98.7 \pm 0.3$ | $\mathbf{93.9} \pm 1.5$ | $95.1 \pm 1.2$ | $\mathbf{89.6} \pm 2.5$ |
| **ElasticNet** | $93.8 \pm 1.5$ | $75.6 \pm 5.3$ | $96.8 \pm 0.8$ | $92.9 \pm 1.7$ | $98.5 \pm 0.4$ | $90.3 \pm 2.3$ | $95.8 \pm 1.0$ | $86.6 \pm 3.1$ |
| **Lasso** | $98.0 \pm 0.5$ | $70.6 \pm 6.2$ | $98.3 \pm 0.4$ | $92.6 \pm 1.8$ | $95.8 \pm 1.0$ | $90.7 \pm 2.2$ | $98.0 \pm 0.5$ | $84.8 \pm 3.5$ |
| **Ridge** | $99.0 \pm 0.2$ | $72.9 \pm 5.8$ | $99.6 \pm 0.1$ | $92.8 \pm 1.7$ | $99.6 \pm 0.1$ | $85.8 \pm 3.3$ | $99.1 \pm 0.2$ | $83.6 \pm 3.7$ |
| **PCR** | $79.2 \pm 4.6$ | $74.4 \pm 5.5$ | $94.7 \pm 1.3$ | $92.3 \pm 1.8$ | $89.6 \pm 2.5$ | $81.3 \pm 4.2$ | $86.5 \pm 3.1$ | $80.5 \pm 4.4$ |
| **PLS** | $97.3 \pm 0.7$ | $52.3 \pm 9.0$ | $96.9 \pm 0.8$ | $93.0 \pm 1.7$ | $97.2 \pm 0.7$ | $67.1 \pm 6.8$ | $94.3 \pm 1.4$ | $84.8 \pm 3.5$ |
| **SHORE** | $82.4 \pm 4.0$ | $73.2 \pm 5.8$ | $87.1 \pm 3.0$ | $92.0 \pm 1.9$ | $76.7 \pm 5.1$ | $51.7 \pm 9.1$ | $89.9 \pm 2.4$ | $78.5 \pm 4.8$ |
| **S-VAE** | $43.1 \pm 10.1$ | $31.8 \pm 11.2$ | $88.2 \pm 2.8$ | $91.4 \pm 2.0$ | $76.7 \pm 5.1$ | $63.7 \pm 7.4$ | $74.1 \pm 5.6$ | $66.8 \pm 6.9$ |

## B.2 TRUE LATENT GRAPH RECOVERY

Here, we analyze the ability of CFA to recover the true latent graph structure when given data that is actually generated from the causal model we assume (i.e., the model in Section 2). Specifically, we evaluate how well our causal clustering step correctly identifies which predictors belong to which mediator and confounder clusters, and which clusters are connected to which responses. We report the F1-score for this cluster assignment task across different sample sizes.

The results, shown in Figure 4, demonstrate that CFA is indeed capable of accurately recovering the true underlying model, and its recovery performance improves rapidly with more data. As shown, even with only 200 samples, CFA achieves an F1-score of 0.92, and it approaches perfect recovery with 300 samples. This provides strong empirical evidence that our method is not just a black-box predictive model but is successfully identifying and leveraging the ground-truth causal structures as intended, when such structures exist.

# C Computational complexity analysis

In this section, we analyze the computational complexity of our proposed CFA method, both theoretically and empirically.

## C.1 Theoretical analysis

The scalability of CFA before the augmented regression is determined by its most computationally intensive stage: Causal Clustering. The complexities of the other pre-processing steps (cluster classification and feature construction) are subsumed by this stage. A naive approach would compute a dense $n_1 \times n_1$ similarity matrix, costing $\mathcal{O}(n_1^2 n_2)$. Our implementation, however, relies on an optimized approach that builds a sparse similarity graph directly. This is highly effective because, in most real-world high-dimensional systems, causal dependencies are localized—a given predictor typically affects only a small subset of responses.

The optimized algorithm proceeds in two main steps:

1. **Dependency identification:** We first perform pairwise dependency tests between all predictors and all responses. This step costs $\mathcal{O}(N n_1 n_2)$ and is embarrassingly parallelizable.

2. **Sparse similarity construction:** For each response $Y_k$, we identify the subset of $d_k$ predictors that are dependent on it. We then only increment the similarity scores for pairs of predictors within this much smaller subset. The total cost for this step across all responses is $\mathcal{O}(\sum_{k=1}^{n_2} d_k^2)$.

Therefore, the total time complexity of CFA is

$$\mathcal{O}\left(N n_1 n_2 + \sum_{k=1}^{n_2} d_k^2\right) + T_{\text{ElasticNet}}(N, n_1 + n_c + n_m, n_2),$$

where $T_{\text{ElasticNet}}(N, n_1 + n_c + n_m, n_2)$ is the cost of the Elastic Net regression with $N$ samples, $n_1 + n_c + n_m$ predictors, and $n_2$ responses. Given that an Elastic Net solver requires $\mathcal{O}(n_2 N(n_1 + n_c + n_m)K)$ time, where $K$ is the number of iterations, the full complexity is

$$\mathcal{O}\left(N n_1 n_2 + \sum_{k=1}^{n_2} d_k^2 + n_2 N(n_1 + n_c + n_m)K\right).$$

This complexity is nearly linear in the number of predictors and avoids the quadratic or cubic scaling bottlenecks. Thus, the CFA framework is computationally scalable to large-scale applications.

## C.2 Empirical analysis

To complement the theoretical analysis, we empirically evaluate the wall-clock time of CFA on synthetic datasets while varying the number of predictors and responses. We also examine how CFA's runtime depends on its additional structural hyperparameters.

**Running time of CFA.** To empirically validate this theoretical analysis, we measured the wall-clock time of our method on synthetic data with $N = 500$ samples while varying the number of predictors ($n_1$) and responses ($n_2$). The results, shown in Table 5, confirm that CFA is highly scalable in practice. For instance, even in a challenging high-dimensional setting with 4,000 predictors and 4,000 responses, our Python implementation completes in just over 1.5 minutes on a standard laptop (MacBook Pro M1), demonstrating its feasibility for large-scale applications.

**Sensitivity to CFA hyperparameters.** Beyond scaling with $(n_1, n_2)$, we also empirically examined how the wall–clock time of CFA depends on its three additional structural hyperparameters: the total number of clusters $n_c + n_m$, the significance level used in the dependence tests for clustering (`clustering_sig`), and the significance level used for cluster classification (`class_sig`). To isolate these effects, we fixed a synthetic dataset generated from our model in Appendix A.2. For each hyperparameter, we varied its value over a wide range while keeping the remaining two at default

Table 5: Empirical wall-clock time (in minutes) of CFA on synthetic data with $N = 500$ samples, varying the number of predictors ($n_1$) and responses ($n_2$). Each entry reports the mean and standard deviation over 20 runs.

| $n_1 \backslash n_2$ | 500 | 1000 | 2000 | 4000 |
|---|---|---|---|---|
| 500 | $0.01 \pm 0.01$ | $0.02 \pm 0.01$ | $0.03 \pm 0.01$ | $0.06 \pm 0.01$ |
| 1000 | $0.03 \pm 0.01$ | $0.05 \pm 0.01$ | $0.09 \pm 0.01$ | $0.20 \pm 0.01$ |
| 2000 | $0.07 \pm 0.01$ | $0.13 \pm 0.01$ | $0.25 \pm 0.02$ | $0.56 \pm 0.05$ |
| 4000 | $0.20 \pm 0.02$ | $0.41 \pm 0.03$ | $0.78 \pm 0.06$ | $1.57 \pm 0.14$ |

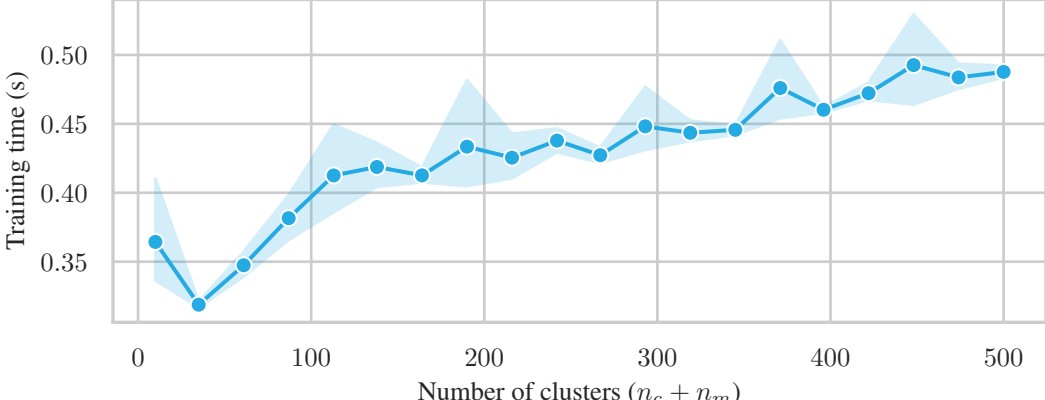

Figure 5: Wall–clock training time of CFA vs. total number of clusters $n_c + n_m$. The shaded area represents the 95% confidence interval over 10 runs.

settings ($m = 10$, `clustering_sig` $= 0.05$, `class_sig` $= 0.01$). For every configuration we ran CFA 10 times and recorded the wall–clock time of the `fit` call.

First, we varied the total number of clusters $m$ from 10 to 500 in 20 approximately evenly spaced values. As shown in Figure 5, the mean runtime increased smoothly from about $0.36$ seconds at $m = 10$ to about $0.49$ seconds at $m = 500$, with relatively small standard deviations across repetitions. This linear growth is consistent with our complexity analysis in Appendix C, where the dependence on $n_c + n_m$ only enters through the clustering stage and the dimensionality of the augmented feature matrix.

Next, we varied the two significance levels over several orders of magnitude, from $10^{-4}$ up to 1. For the clustering threshold (clustering_sig_level), the mean runtimes remained in a narrow band (roughly $0.38$–$0.47$ seconds) with no systematic trend as the threshold changed (Figure 6). A similar behavior was observed for the cluster-classification threshold (cluster_class_sig_level), where runtimes again stayed within a tight range (Figure 7). This is expected: these thresholds only affect which edges are retained in the similarity graph and the subsequent labels of clusters, but they do not change the asymptotic cost of the similarity construction or the final regression.

Overall, these experiments confirm that (i) runtime scales approximately linearly with the total number of clusters in practice, in line with the theoretical analysis, and (ii) CFA's wall–clock time is essentially insensitive to the exact choice of the two $p$-value thresholds within the ranges we consider. In particular, the additional structural hyperparameters do not introduce any hidden exponential or super–linear cost beyond what is already captured by the $(N, n_1, n_2, n_c + n_m)$ scaling analyzed in Appendix C.

### C.3 RUNTIME COMPARISON WITH BASELINES

To complement the CFA-only scaling results in Appendix C, we also compare the wall–clock time of CFA against all baselines on synthetic data as a function of the number of predictors and responses.

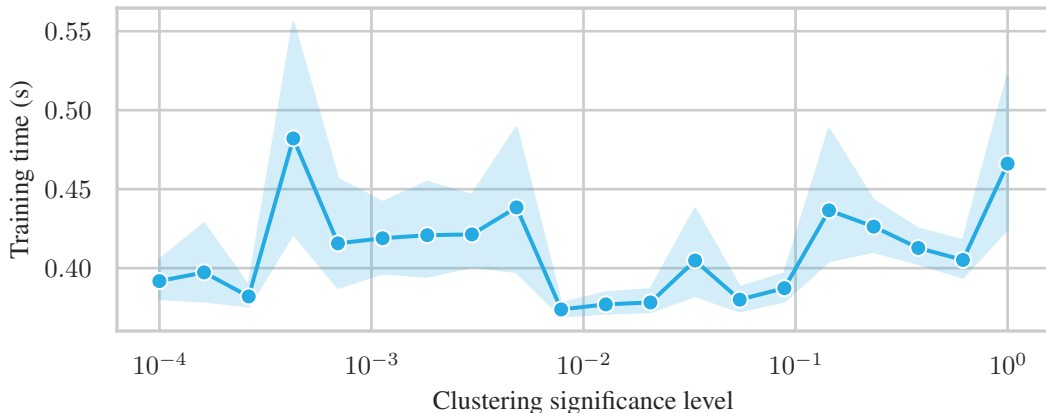

Figure 6: Wall–clock training time of CFA vs. clustering significance level (`clustering_sig`). The shaded area represents the 95% confidence interval over 10 runs.

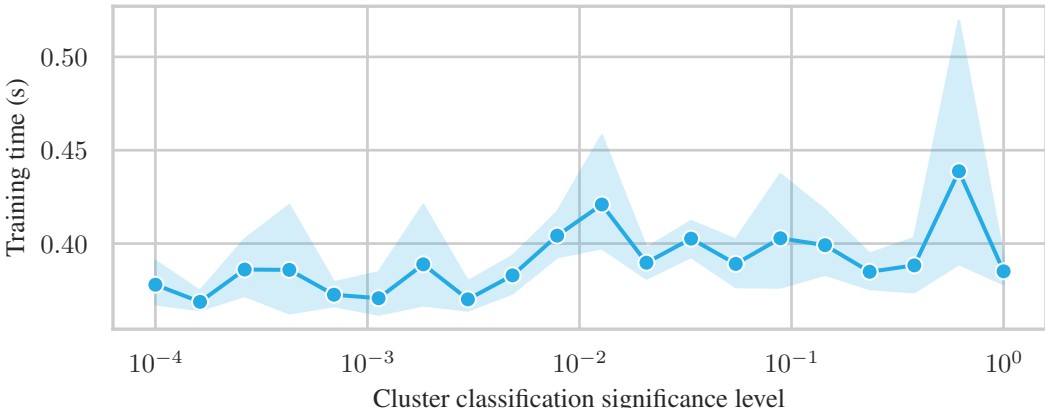

Figure 7: Wall–clock training time of CFA vs. cluster-classification significance level (`class_sig`). The shaded area represents the 95% confidence interval over 10 runs.

We generate data from the model in Appendix A.2 with $N = 500$ samples and vary the number of predictors and responses jointly as $n_1 = n_2 \in \{200, 500, 1000, 1500, 2000\}$. For each configuration, we draw a fresh dataset and fit the following methods: CFA, `ElasticNet`, `Lasso`, `Ridge`, Principal Component Regression (`PCR`), Partial Least Squares (`PLS`), Supervised VAE (`S-VAE`), and `SHORE`. All methods are run with a fixed set of hyperparameters (consistent with Appendix A.1), and we record the wall–clock time of a single training run (i.e., `fit` call). Each setting $(n_1, n_2)$ is repeated 10 times with different random seeds, and we report the mean runtime in Figure 8.

As seen in Figure 8, very simple closed-form or nearly closed-form methods such as Ridge is the fastest, staying well below 0.1 s even at $n_1 = n_2 = 2000$. CFA is slower than these but scales gracefully: its mean runtime grows from about 0.10 s at $n_1 = n_2 = 200$ to about 7–8 s at $n_1 = n_2 = 2000$, in line with the near-linear scaling in $(n_1, n_2)$ observed in Table 5. Crucially, CFA is *much faster* than the strong sparse baselines ElasticNet and Lasso in this regime. For example, at $n_1 = n_2 = 2000$, CFA fits in roughly 8 s, whereas ElasticNet and Lasso require approximately 40–50 s—about a 5×–6× slowdown compared to CFA.

This gap is explained by the different computational bottlenecks. The ElasticNet baseline solves one large, high-dimensional sparse regression directly on the original features: for each response, coordinate descent must repeatedly sweep over all $n_1$ highly correlated predictors until convergence. As $n_1 = n_2$ grows, this "flat" optimization problem becomes increasingly heavy, so runtime grows quickly.

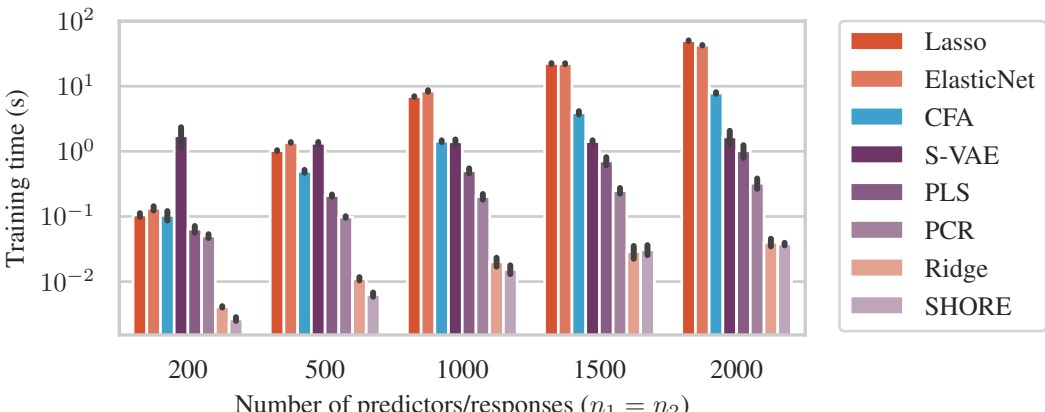

Figure 8: Log wall-clock training time comparison of CFA against baselines on synthetic data, varying the number of predictors and responses jointly as $n_1 = n_2$. Error bar indicates standard deviation over 10 runs.

CFA pays a modest extra cost up front to organize the predictors. The causal–clustering stage builds a similarity graph, forms mediator/confounder clusters, and constructs augmented features $[\mathbf{X}, \mathbf{Z}^{(m)}, \mathbf{Z}^{(c)}]$ that concentrate shared variation into a smaller number of information–rich directions. The downstream ElasticNet in CFA therefore sees a simpler effective problem (less entangled, better–structured features, and sparser direct effects), and converges in far fewer iterations than plain ElasticNet on the raw $\mathbf{X}$. This is why, despite its additional stages, CFA is several times faster in wall–clock time than the ElasticNet baseline in the high-dimensional regime.

Overall, these experiments show that CFA is competitive in wall–clock time: it is only modestly more expensive than very fast closed-form methods, yet substantially *more efficient* than strong sparse baselines such as ElasticNet and Lasso, while delivering the accuracy gains reported in Section 4.

## D  PROOFS

**Theorem 3.1** (Dependence via causal graph structures). *Under the generative model described in Section 2, for any distinct pair of predictors $(X_i, X_j)$, and any response variable $Y_k$, we have:*

$$X_i \not\perp\!\!\!\perp Y_k \text{ and } X_j \not\perp\!\!\!\perp Y_k \quad \Longleftrightarrow \quad \mathbb{G}_{ijk} \text{ is non-empty.}$$

**Remark D.1.** *The proof of Theorem 3.1 relies on the notion of* d-separation *(Pearl, 2009), which characterizes conditional independence relations in a causal graph. In general, equivalence between d-separation and conditional independence requires the assumption of* faithfulness*. However, for our purposes, we only require the milder assumption that marginal independence between any pair of variables implies d-separation (i.e., no active paths between them).*

*Proof.* We begin with the following lemma.

**Lemma D.1.** *Under the causal generative model in Section 2, a variable $X_i$ is dependent on $Y_k$ (i.e., $X_i \not\perp\!\!\!\perp Y_k$) if and only if they share at least one common ancestor[5]. This ancestor may be $X_i$ itself or an upstream latent variable.*

This lemma follows from the standard d-separation criterion in causal graphical models (see, e.g., Pearl (2009)). We now prove the theorem by enumerating the five cluster configurations in Table 1 and applying Lemma D.1 to characterize all possible structures in which both $X_i$ and $X_j$ are dependent on $Y_k$.

---

[5]A variable $A$ is an ancestor of variable $B$ if there exists a directed path from $A$ to $B$.

**Case (i):** $X_i, X_j \in \mathcal{I}_t^{(m)}$. There are two graph structures shown in Table 1: (1) Both $X_i$ and $X_j$ directly causing $Y_k$, and (2) both cause a shared mediator $Z_t^{(m)}$, which then cause $Y_k$. In either case, both $X_i$ and $X_j$ are ancestors of $Y_k$, thus Lemma D.1 implies they are each dependent on $Y_k$.

For the converse, suppose both $X_i$ and $X_j$ are dependent on $Y_k$. Lemma D.1 implies they both have a common ancestor with $Y_k$. Since the only ancestors of both $X_i$ and $X_j$ are themselves, they both should be ancestors of $Y_k$. Hence, either both have direct edges to $Y_k$, or the mediator $Z_t^{(m)}$ is connected to $Y_k$, which are the structures shown in the table.

**Case (ii):** $X_i, X_j \in \mathcal{I}_t^{(c)}$. Table 1 lists a dashed box around $Z_t^{(c)}$ and its children, with a dashed edge to $Y_k$ indicating that at least one of them causes $Y_k$. If any such connection exists, then $Z_t^{(c)}$ becomes a common ancestor of both $X_i$ and $Y_k$, and similarly of $X_j$ and $Y_k$, ensuring both dependencies by Lemma D.1.

Conversely, suppose $X_i \not\perp\!\!\!\perp Y_k$. If $X_i$ is not directly connected to $Y_k$, it must be d-connected via a common ancestor. Since its only ancestor is $Z_t^{(c)}$, either $Z_t^{(c)}$ or one of its other children must cause $Y_k$, again implying one of the structures shown in the table. The same argument applies symmetrically to $X_j$.

**Case (iii):** $X_i \in \mathcal{I}_{t_1}^{(m)}$, $X_j \in \mathcal{I}_{t_2}^{(m)}$, **with** $t_1 \neq t_2$. Table 1 shows the structures where each of $X_i$ and $X_j$ (or their corresponding mediators) causes $Y_k$. In these cases, both predictors are ancestors of $Y_k$, so dependence follows by Lemma D.1.

Conversely, if both $X_i$ and $X_j$ are dependent on $Y_k$, they must each be ancestors of $Y_k$, as neither has any other ancestors. This occurs if either they directly cause $Y_k$ or their respective mediators do—matching the graphs in the table.

**Case (iv):** $X_i \in \mathcal{I}_{t_1}^{(c)}$, $X_j \in \mathcal{I}_{t_2}^{(c)}$, **with** $t_1 \neq t_2$. This case follows directly from the reasoning in Case (ii), applied separately to $X_i$ and $X_j$ and their respective confounders $Z_{t_1}^{(c)}$ and $Z_{t_2}^{(c)}$. Since these confounders are disjoint, both dependencies occur if and only if the structures in the table hold.

**Case (v):** $X_i \in \mathcal{I}_{t_1}^{(m)}$, $X_j \in \mathcal{I}_{t_2}^{(c)}$. This case directly combines the arguments from Cases (iii) and (iv), applied respectively to $X_i$ and $X_j$. Each must satisfy the structural conditions identified in their corresponding cases for both to be dependent on $Y_k$. Hence, the structures shown in the table are both necessary and sufficient. $\square$

**Theorem 3.2** (Probabilistic justification of the similarity metric). *Under the edge-generation model of Assumption 1, Table 1 lists the probabilities of $\mathbb{G}_{ijk}$ being non-empty conditioned on each of the five cases of the table. These probabilities, which are a consequence of Assumption 1, imply*

$$\mathbb{P}\left(|\mathbb{G}_{ijk}| > 0 \mid X_i, X_j \text{ in the same cluster}\right) > \mathbb{P}\left(|\mathbb{G}_{ijk}| > 0 \mid X_i, X_j \text{ in different clusters}\right),$$

*where $|\mathbb{G}_{ijk}|$ denotes the cardinality of the set of causal subgraphs compatible with the underlying model of Section 2.*

*Proof.* We first compute the exact probabilities in each of the five valid cases in Table 1.

**Same-cluster cases:** (i) If $X_i, X_j \in \mathcal{I}_t^{(m)}$, the graphs in $\mathbb{G}_{ijk}$ arise either from edges $Z_t^{(m)} \to Y_k$ or both $X_i \to Y_k$ and $X_j \to Y_k$. These events are independent with probabilities $p_{my}$ and $p_{xy}^2$, and their intersection occurs with probability $p_{my} \cdot p_{xy}^2$. Thus

$$\mathbb{P}(|\mathbb{G}_{ijk}| > 0 \mid X_i, X_j \in \mathcal{I}_t^{(m)}) = p_{my} + p_{xy}^2 - p_{my}p_{xy}^2 = p_{my} + (1 - p_{my})p_{xy}^2.$$

(ii) If $X_i, X_j \in \mathcal{I}_t^{(c)}$, both share the same latent confounder $Z_t^{(c)}$. The response $Y_k$ must be connected either to this confounder (with probability $p_{cy}$) or to any of the $m_c$ predictors in the cluster (each connected with probability $p_{xy}$). The probability that all these fail is $(1 - p_{cy})(1 - p_{xy})^{m_c}$, hence

$$\mathbb{P}(|\mathbb{G}_{ijk}| > 0 \mid X_i, X_j \in \mathcal{I}_t^{(c)}) = 1 - (1 - p_{cy})(1 - p_{xy})^{m_c}.$$

**Different-cluster cases:** **(iii)** If $X_i \in \mathcal{I}_{t_1}^{(m)}$, $X_j \in \mathcal{I}_{t_2}^{(m)}$, $t_1 \neq t_2$, the graphs arise when at least one of $\{X_i, Z_{t_1}^{(m)}\}$ and one of $\{X_j, Z_{t_2}^{(m)}\}$ connect to $Y_k$. These two events are independent, and each of them occurs with probability $1 - (1 - p_{xy})(1 - p_{my})$, yielding

$$\mathbb{P}(|\mathbb{G}_{ijk}| > 0 \mid X_i \in \mathcal{I}_{t_1}^{(m)}, \, X_j \in \mathcal{I}_{t_2}^{(m)}) = (1 - (1 - p_{xy})(1 - p_{my}))^2 \, .$$

**(iv)** If $X_i \in \mathcal{I}_{t_1}^{(c)}$, $X_j \in \mathcal{I}_{t_2}^{(c)}$ with $t_1 \neq t_2$, the causal structures connecting $X_i$ and $X_j$ to $Y_k$ are independent. For each confounder cluster, dependence arises if either the confounder itself or one of its children (i.e., the variables in the corresponding cluster) is connected to $Y_k$. The probability that none of these are connected is $(1 - p_{cy})(1 - p_{xy})^{m_c}$. Thus, the probability that the graph $\mathbb{G}_{ijk}$ is not empty is

$$\mathbb{P}(|\mathbb{G}_{ijk}| > 0 \mid X_i \in \mathcal{I}_{t_1}^{(c)}, \, X_j \in \mathcal{I}_{t_2}^{(c)}) = (1 - (1 - p_{cy})(1 - p_{xy})^{m_c})^2 \, .$$

**(v)** If $X_i \in \mathcal{I}_{t_1}^{(m)}$, $X_j \in \mathcal{I}_{t_2}^{(c)}$, the event $|\mathbb{G}_{ijk}| > 0$ occurs if both blocks $\mathcal{I}_{t_1}^{(m)}$ and $\mathcal{I}_{t_2}^{(c)}$ induce dependence with $Y_k$. For the mediator-side, dependence arises from either $X_i$ or $Z_{t_1}^{(m)}$ connecting to $Y_k$, which happens with probability $1 - (1 - p_{xy})(1 - p_{my})$. For the confounder-side, the same logic as in Case (iv) applies, giving probability $1 - (1 - p_{cy})(1 - p_{xy})^{m_c}$. Since these two sides are independent, the probability that the graph $\mathbb{G}_{ijk}$ is not empty is

$$\mathbb{P}(|\mathbb{G}_{ijk}| > 0 \mid X_i \in \mathcal{I}_{t_1}^{(m)}, \, X_j \in \mathcal{I}_{t_2}^{(c)}) = (1 - (1 - p_{xy})(1 - p_{my})) \left(1 - (1 - p_{cy})(1 - p_{xy})^{m_c}\right) \, .$$

We now derive the approximations listed in Table 1, using the assumption that all edge probabilities $p_{xy}, p_{my}, p_{cy} \ll 1$, and retaining only first- or second-order terms as appropriate:

**(i)** $\mathbb{P}(|\mathbb{G}_{ijk}| > 0 \mid X_i, X_j \in \mathcal{I}_t^{(m)}) = p_{my} + (1 - p_{my})p_{xy}^2 \approx p_{my}$, since $p_{xy}^2$ is second order and negligible compared to $p_{my}$.

**(ii)** $\mathbb{P}(|\mathbb{G}_{ijk}| > 0 \mid X_i, X_j \in \mathcal{I}_t^{(c)}) = 1 - (1 - p_{cy})(1 - p_{xy})^{m_c}$. Expanding and discarding second-order terms like $p_{cy}p_{xy}$ and $p_{xy}^k$ for $k \geq 2$, we get the approximation

$$\approx p_{cy} + m_c p_{xy}.$$

**(iii)** $\mathbb{P}(|\mathbb{G}_{ijk}| > 0 \mid X_i \in \mathcal{I}_{t_1}^{(m)}, X_j \in \mathcal{I}_{t_2}^{(m)}) = (1 - (1 - p_{xy})(1 - p_{my}))^2 \approx (p_{xy} + p_{my})^2$.

**(iv)** $\mathbb{P}(|\mathbb{G}_{ijk}| > 0 \mid X_i \in \mathcal{I}_{t_1}^{(c)}, X_j \in \mathcal{I}_{t_2}^{(c)}) = (1 - (1 - p_{cy})(1 - p_{xy})^{m_c})^2$. As in (ii), we approximate it by

$$\approx (p_{cy} + m_c p_{xy})^2.$$

**(v)** $\mathbb{P}(|\mathbb{G}_{ijk}| > 0 \mid X_i \in \mathcal{I}_{t_1}^{(m)}, X_j \in \mathcal{I}_{t_2}^{(c)}) = (1 - (1 - p_{xy})(1 - p_{my})) \times (1 - (1 - p_{cy})(1 - p_{xy})^{m_c})$, which we approximate as

$$\approx (p_{xy} + p_{my})(p_{cy} + m_c p_{xy}).$$

We now complete the proof using the calculated probabilities. Let $\pi_c \in [0, 1]$ denote the prior probability that a given predictor $X_j$ belongs to a confounder cluster; thus, the prior probability of belonging to a mediator cluster is $\pi_m = 1 - \pi_c$. Note that the assignments of variables to clusters are independent. Conditioned on same-cluster membership, the probability that $(X_i, X_j)$ belong to a mediator cluster is

$$\mathbb{P}(\text{same mediator cluster} \mid \text{same cluster}) = \frac{\pi_m^2}{\pi_m^2 + \pi_c^2},$$

and similarly for confounder clusters:

$$\mathbb{P}(\text{same confounder cluster} \mid \text{same cluster}) = \frac{\pi_c^2}{\pi_m^2 + \pi_c^2}.$$

Therefore,

$$\mathbb{P}(|\mathbb{G}_{ijk}| > 0 \mid X_i, X_j \text{ in same cluster}) = \frac{\pi_m^2}{\pi_m^2 + \pi_c^2} \cdot P_{\text{mm}}^{\text{same}} + \frac{\pi_c^2}{\pi_m^2 + \pi_c^2} \cdot P_{\text{cc}}^{\text{same}},$$

where

$$P_{\text{mm}}^{\text{same}} := \mathbb{P}(|\mathbb{G}_{ijk}| > 0 \mid X_i, X_j \in \mathcal{I}_t^{(m)}), \quad P_{\text{cc}}^{\text{same}} := \mathbb{P}(|\mathbb{G}_{ijk}| > 0 \mid X_i, X_j \in \mathcal{I}_t^{(c)}).$$

For different-cluster pairs, the three configurations occur with probabilities:

$$\mathbb{P}(\text{mm-diff}) = \pi_m^2, \quad \mathbb{P}(\text{cc-diff}) = \pi_c^2, \quad \mathbb{P}(\text{mc-diff}) = 2\pi_m\pi_c.$$

Hence,

$$\mathbb{P}(|\mathbb{G}_{ijk}| > 0 \mid X_i, X_j \text{ in different clusters}) = \pi_m^2 \cdot P_{\text{mm}}^{\text{diff}} + \pi_c^2 \cdot P_{\text{cc}}^{\text{diff}} + 2\pi_m\pi_c \cdot P_{\text{mc}}^{\text{diff}}.$$

Substituting the approximations:

$$P_{\text{mm}}^{\text{same}} \approx p_{my},$$
$$P_{\text{cc}}^{\text{same}} \approx p_{cy} + m_c p_{xy},$$
$$P_{\text{mm}}^{\text{diff}} \approx (p_{my} + p_{xy})^2,$$
$$P_{\text{cc}}^{\text{diff}} \approx (p_{cy} + m_c p_{xy})^2,$$
$$P_{\text{mc}}^{\text{diff}} \approx (p_{my} + p_{xy})(p_{cy} + m_c p_{xy}),$$

we conclude:

$$\mathbb{P}(|\mathbb{G}_{ijk}| > 0 \mid X_i, X_j \text{ in same cluster}) \approx \frac{\pi_m^2}{\pi_m^2 + \pi_c^2} p_{my} + \frac{\pi_c^2}{\pi_m^2 + \pi_c^2} (p_{cy} + m_c p_{xy}),$$

$$\mathbb{P}(|\mathbb{G}_{ijk}| > 0 \mid X_i, X_j \text{ in different clusters}) \approx \pi_m^2 (p_{my} + p_{xy})^2 + \pi_c^2 (p_{cy} + m_c p_{xy})^2$$
$$+ 2\pi_m\pi_c (p_{my} + p_{xy})(p_{cy} + m_c p_{xy}) = (\pi_m(p_{my} + p_{xy}) + \pi_c(p_{cy} + m_c p_{xy}))^2.$$

Since the first probability is a first-order quantity and the second one is second-order in small parameters, for sufficiently small $p_{xy}, p_{my}, p_{cy}$, the inequality holds, completing the proof. $\qquad\square$

