# OpenReview forum: "CFA: Causal Feature Augmentation for High-Dimensional Linear Regression"
_ICLR.cc/2026/Conference — Submitted to ICLR 2026_

### Official Review · Reviewer_eEeh · 2025-10-25

**Soundness:** 2
**Presentation:** 2
**Contribution:** 2
**Rating:** 2
**Confidence:** 4

**Summary:**

The authors introduce Causal Feature Augmentation (CFA), a framework for high-dimensional linear regression. The key idea is to consider the effect of mediators and confounders that affect the relationship between X and Y. The authors propose an algorithm to separate the predictors based on pairwise similarity (how to predictors jointly affect the response) and construct augmented confounding and mediation features. The augmented features [X,Z_c,Z_m] are then fitted with an Elastic net as a standard regression problem with L1 and L2 penalties. CFA shows better correlation results compared to those commonly used high-dimensional regression methods.

**Strengths:**

1.	Theoretical analysis, especially the explanation of why counting shared Y-dependencies can reveal latent grouping structure, is solid.
2.	The performance gain is pretty big especially under the scenarios with small sample size.
3.	The consideration of The mediator/ confounder effect is pretty intuitive, and the features construction is also straightforward.

**Weaknesses:**

1.	It seems that the similarity matrix replies on a quite number of dependency test. This will be quite inefficient especially when dimension of X and Y are large.
2.	The performance on real data (S&P 500) shows some improvement, but the correlation is low on test set, which makes little practical use.
3.	Recovering each confounder using only the first principal component per cluster can be quite restrictive. Sometimes the true shared variability can be multi factors, which may require more PCs.

**Questions:**

1.	Could the author provide results on more commonly used regression datasets? e.g., from UCI Machine Learning Repository.
2.	How would Z_m Z_c be sensitive to those important hyper parameters, such as he number of clusters, p-value thresholds etc.
3.	The sample sizes in all experiments are very small, which does fit the many problems in modern data science. The authors should demonstrate clear under what scenarios this method can be useful. Perhaps a real dateset (e.g., a rare disease) with limited sample size may help.
4.	Some important baselines are missed, such as group lasso, multi-task lasso. More modern AI methods based on representation learning shall also be considered, such as a fully-connect network.
5.	Detailed scalability experiments are needed for different sample size and different dimension of X,Y.

---

> ### Author Response · Authors · 2025-11-21
>
> We thank the reviewer for reading our paper thoroughly and giving us feedback. We respond to all of their comments and questions below.
>
> ## W1
> The similarity construction indeed involves many dependency tests, but its complexity is controlled. As detailed in Appendix C, building the dependency structure requires $\mathcal{O}(N n_1 n_2)$ operations, which is on the same order as computing all $X$–$Y$ covariances or performing a full pass of a multi-output linear regression. This step is also embarrassingly parallelizable.
>
> Empirically, Appendix C.2 shows that even in a high-dimensional setting with $n_1 = n_2 = 4000$ and $N = 500$, our Python implementation of CFA runs in about **1.6 minutes** on a MacBook Pro M1, indicating that the method is practical at scale.
>
> Moreover, new experiments (see response to Reviewer DDDW) in Appendix C.3 show that **CFA is 5-6 times faster than Elastic Net** on a dataset with $N=500$ and $n_1=n_2=2000$. This is because CFA's information-rich features help its regression problem converge much faster than that of naive regression on $\mathbf{X}$ (i.e., Elastic Net).
>
> ## W2
> We agree that the raw test correlations on S&P 500 look small in absolute terms, but this is expected in this domain. As noted on line 483, predicting next-day stock returns is notoriously difficult due to market efficiency and high noise; in this literature, low single-digit out-of-sample correlations are already considered meaningful. In Table 2, CFA consistently achieves the highest test correlation across all rolling windows, with smaller train–test gaps than strong baselines, so we view the gains as practically relevant despite the low absolute values.
>
> ## W3
>  We agree that latent confounders can be multi-dimensional. While our framework can technically be extended to include top-$k$ PCs, we deliberately restricted the default implementation to the first PC as a specific design choice for the low-sample regime ($N \ll P$) that our paper targets.
>
>  In data-scarce settings, estimating higher-order principal components often captures noise rather than true signal (instability). By restricting the feature construction to the single dominant direction of variation (the 1st PC), we effectively impose a strong regularization that prioritizes estimation robustness over expressiveness. We found this trade-off essential for preventing overfitting.
>
> ## Q1
> Most commonly used UCI regression benchmarks are **single-output** (scalar $Y$), whereas CFA is specifically designed for **high-dimensional multi-response** regression. Applying it to 1D targets would not really test its core contribution. Instead, we evaluate on multi-output benchmarks from Mulan (ATP-7D, OES-10, RF-2, SCM-1D) and on the S&P 500 multi-stock return prediction task, which are much closer to the setting CFA is intended for. We refer the reviewer to new Section 4.4 and Appendix A.
>
> ## Q2
> The constructions of $Z^{(m)}$ and $Z^{(c)}$ indeed depend on the clustering and testing hyperparameters (number of clusters, $p$-value thresholds), but in practice we found CFA to be reasonably robust to these choices. All such hyperparameters are tuned via cross-validation (using TPE), and within a broad range (as explained in Appendix A), validation performance changes smoothly rather than abruptly.
>
> Intuitively, if we choose too few clusters or very strict $p$-value thresholds, most structure is absorbed into a small number of features and CFA behaves similarly to Elastic Net on $X$ alone. If we choose too many clusters or very lax thresholds, we obtain more constructed features, but unhelpful ones are shrunk by the Elastic Net penalty. In both cases, the augmented regression step mitigates sensitivity by down-weighting poorly chosen $Z^{(m)}$ and $Z^{(c)}$.
>
> ## Q3
> CFA is explicitly designed for high-dimensional, low-sample regimes where $n_1, n_2$ are large relative to $N$. Our experiments are chosen to reflect this: e.g., in the synthetic setup we use $n_1 = n_2 = 1000$ with $N$ as small as 100, and in S&P 500 we have roughly 500 stocks (predictors/responses) with $N \approx 700$ time points. The Mulan datasets (ATP-7D, OES-10, RF-2, SCM-1D) in Appendix B are also multi-output with moderately small $N$ relative to the number of variables.
>
> These are precisely the scenarios where we observe the largest gains: many correlated predictors and responses, but limited samples (e.g., finance, environmental monitoring, multi-output forecasting, and potentially rare-disease or other biomedical settings). In large-sample regimes, CFA naturally reduces to behaving similarly to a strong regularized baseline such as Elastic Net.

---

> ### Author Response · Authors · 2025-11-21
>
> (response continued)
>
> ## Q4
> We agree that structured sparsity and representation learning are important baselines, but multi-task group lasso formulations are somewhat misaligned with our setting. In particular, standard multi-task lasso imposes _row-sparsity_ on the coefficient matrix, effectively forcing the same subset of predictors $X$ to be either active or inactive **across all responses** (see [here](https://scikit-learn.org/stable/modules/linear_model.html#multi-task-lasso). In real-world applications (e.g., S&P 500, Mulan datasets), different responses naturally depend on different subsets of predictors, so such shared-support assumptions are overly restrictive and can hurt performance; this is precisely why we build CFA to allow task-specific patterns via augmented features and per-response sparsity. Indeed, our Cluster Regressor (CR) baseline effectively tests this grouping logic without the joint-support constraint and is actually a more flexible and sophisticated version of Group/Multi-task Lasso, yet it still underperformed CFA (Figure 3/Table 2), proving that simply grouping variables is insufficient compared to extracting latent features
>
> On the “modern AI” side, we already include representation-learning baselines: S-VAE (a supervised deep generative model) and SHORE (Li et al., 2024), along with PCR and PLS. These methods underperform CFA in the low-sample, high-dimensional regime we target. Since CFA ultimately solves a _linear_ regression on an augmented feature space, the most direct and robust baseline is a strong regularized linear model—Elastic Net—which we do compare against.
>
> ## Q5
> We have provided a detailed theoretical complexity and empirical scalability analysis in Appendix C, which reports empirical wall-clock times while varying predictor and response dimensions from 500 up to 4,000. The results demonstrate that CFA is highly efficient, completing a $4000 \times 4000$ dimensional task in under 1.6 minutes on a standard laptop.
>
> Additionally, we have conducted new experiments on the sensitivity of the running time to hyperparameters in Appendix C.2. These experiments confirm that (i) runtime scales approximately linearly with the total number of clusters in practice, in line with the theoretical analysis, and (ii) CFA’s wall–clock time is essentially insensitive to the exact choice of the two $p$-value thresholds within the ranges we consider.
>
> We also performed new experiments on comparing the running time of CFA against all baselines, detailed in new Appendix C.3 (and also our response to Reviewer DDDW). Briefly, we observed that **CFA is 5-6 times faster than Elastic Net** on a dataset with $N=500$ and $n_1=n_2=2000$. This is because CFA's information-rich features help its regression problem converge much faster than that of naive regression on $\mathbf{X}$ (i.e., Elastic Net).
>
> Regarding sample size ($N$): Our theoretical analysis (Appendix C.1) confirms the complexity is linear in $N$. Since our method targets the "Low-Sample, High-Dimension" regime (where $N \ll n_1$), the computational cost is dominated by the feature dimensions, not the sample count. Thus, the method remains computationally inexpensive even as $N$ scales.

---

### Official Review · Reviewer_4Jaf · 2025-10-31

**Soundness:** 3
**Presentation:** 2
**Contribution:** 2
**Rating:** 4
**Confidence:** 3

**Summary:**

This paper addresses high-dimensional, small-sample linear regression tasks by introducing a method inspired by causal inference concepts. The authors cluster predictors into groups that share latent mechanisms (either confounder-based or mediator-based) and use these latent variables to improve regression performance. The paper defines a similarity metric between predictors, classifies clusters based on intra-cluster correlations, and constructs latent variables either by averaging (for confounder clusters) or via PCA (for mediator clusters). The final regression model is trained with Elastic Net. Experiments on synthetic datasets and S&P 500 stock data demonstrate gains.

**Strengths:**

1. The paper presents a clear motivation and a coherent framework.
2. The proposed distinction between confounder and mediator clusters is intuitively appealing and nicely connected to causal concepts.

**Weaknesses:**

1. Limited novelty: The causal framing adds interpretability but little methodological innovation.
2. Sparse and outdated citations: The reference list includes fewer than 30 works, most from before 2010, raising concerns that relevant modern literature was overlooked.
3. Unconvincing experiments: Synthetic data favor the proposed structure by design, while the S&P 500 results are very weak.

**Questions:**

1. Could the authors clarify whether the method’s advantage persists when the true latent structure does not align with the causal assumptions?
2. Why were the four additional datasets mentioned in the appendix not included in the main paper? They seem to perform better than S&P 500.
3. Have the authors compared their method to modern alternatives? Did the authors conduct a detailed investigation of this field and compare with more advanced baselines?

---

> ### Author Response · Authors · 2025-11-21
>
> We thank the reviewer for their constructive feedback. We respond to all their comments and questions below.
>
> ## W1
> We respectfully disagree that the contribution is only interpretive. Methodologically, it introduces:
> 1. **A new similarity metric with analysis:** The similarity $s(i,j)$ is defined via shared $Y$-dependencies, rather than correlations in $X$-space. Theorems 3.1 and 3.2 show that, under our generative model, this metric preferentially groups predictors sharing latent mediators or confounders, which is what drives the clustering step.
> 2. **Mediator-/confounder-specific feature construction:** Using intra-cluster correlations, clusters are classified as mediator or confounder type, and we build different features for each, yielding a concrete augmentation pipeline that differs from standard clustering, representation learning, or multi-task regularization methods.
> 3. **An integrated regression scheme:** These causality-inspired features are used jointly with $X$ in a single Elastic Net, which consistently improves small-sample performance over strong baselines on both synthetic and five real-world datasets.
>
> So the causal framing leads to a specific, theoretically backed algorithmic design, not just an interpretive rephrasing of existing methods.
>
> ## W2
> Many foundational results for linear and multi-output regression date back before 2010, so it is natural that several core references are older. That said, we have taken care to include recent literature relevant to this specific high-dimensional, low-sample regime. The introduction cites modern methods such as adaptive reduced-rank regression, task clustering, Supervised VAEs, and the very recent SHORE (2024).
>
> To further address your concern and capture the broader modern landscape, we have updated the manuscript to include Xu et al. (2019), which provides a comprehensive survey of the multi-output learning field, and Bertsimas & Van Parys (2017), which advances exact high-dimensional sparse regression via modern optimization techniques, in the related works part of the introduction.
>
> If the reviewer has particular contemporary works in mind that fit our setting, we would be happy to discuss and compare to them as well.
>
> ## W3
> The synthetic experiment is intentionally aligned with our generative model in order to test whether CFA can in fact exploit the mediator/confounder structure it is designed for; this is a standard sanity check rather than our only evidence.
>
> For S&P 500, as noted on line 488, predicting next-day stock returns is a notoriously hard task due to market efficiency and noise. In this setting, low single-digit out-of-sample correlations (2–3% in Table 2) are considered meaningful, and CFA consistently achieves the best test correlation than strong baselines, which we view as practically relevant.
>
> Finally, beyond S&P 500, Appendix B reports results on four additional real-world datasets (ATP-7D, OES-10, RF-2, SCM-1D), where CFA again achieves the highest test correlations across all cases (Table 4), indicating that the empirical gains are not confined to the synthetic setting.

---

> ### Author Response · Authors · 2025-11-21
>
> (response continued)
>
>
> ## Q1
> Yes. In real-world settings the true latent structure almost certainly does _not_ match our idealized mediator/confounder model, yet CFA still shows consistent gains. On S&P 500 and the four additional real datasets in Appendix B, CFA outperforms strong baselines in the low-sample regime, despite misspecification.
>
> Moreover, when the augmented features are not informative, the Elastic Net penalty in Eq. (5) shrinks their coefficients toward zero, so the method effectively falls back to a standard regularized regression. In the synthetic experiment (Figure 3), as $N$ grows large ($N \gg n_1, n_2$), all methods—including CFA and the baselines—converge to similar near-perfect performance, while the main advantage of CFA appears in the small-sample regimes.
>
> ## Q2
> We chose S&P 500 for the main text because it is a well-known, challenging benchmark where meaningful signal is hard to extract, so even small test correlations are informative. As stated on line 364, _“for the sake of space, we include additional experiments on four real-world datasets … in Appendix B.”_ Those four datasets (ATP-7D, OES-10, RF-2, SCM-1D) are fully reported in Appendix B and indeed show even stronger relative gains for CFA (Table 4). To make this clearer, have moved a summary of the four additional real-world experiments (ATP-7D, OES-10, RF-2, SCM-1D) into the main text (new Section 4.4) and keep detailed results in the appendix.
>
> ## Q3
> Yes. We compared CFA against several modern alternatives, not just classical linear models. In particular, we include:
> - **SHORE** (Li et al., 2024), which to the best of our knowledge is the most recent directly relevant method for high-dimensional multi-output regression,
> - **Supervised VAE (S-VAE)** as a deep representation-learning baseline,
> - **PCR** and **PLS** as latent-factor baselines,
> - and strong regularized models (**ElasticNet, Lasso, Ridge**) plus a clustering-based regressor (CR).
>
>
> SHORE is specifically designed for settings with sparse outputs, an assumption that does not hold in our financial and multi-output regression datasets. Consistent with this mismatch, SHORE underperforms across our experiments, which we see as further evidence that CFA’s design is better suited to the dense, low-sample regimes we study.
>
> If the reviewer is aware of additional recent methods tailored to this specific setting (high-dimensional, low-sample, multi-response regression), we would be happy to include and compare against them.

---

### Official Review · Reviewer_DDDW · 2025-11-01

**Soundness:** 2
**Presentation:** 2
**Contribution:** 2
**Rating:** 2
**Confidence:** 3

**Summary:**

The paper proposes Causal Feature Augmentation (CFA), a pragmatic pipeline that engineers a small set of proxy mediator and confounder features from the original predictors and appends them to a standard linear model (Elastic Net) for better multi-response prediction. Concretely, CFA (i) detects which predictors relate to which responses via thresholded two-tailed correlation t-tests, (ii) builds a predictor–predictor similarity score $s(i,j)$ from how often two predictors co-associate with the same responses, then (iii) clusters predictors using bottom-up hierarchical clustering (average linkage). Within each cluster, CFA labels mediator-type vs confounder-type via intra-cluster correlation tests. Augmented features are then constructed to enhance model prediction. Experiments include synthetic linear settings matching the design assumptions and a real S&P 500 prediction task; both show improvements over a plain Elastic Net and several ablations of the augmentation choices.

**Strengths:**

1. The proposed method slots into standard Elastic Net workflows; no bespoke learner or complex training loop is required.
2. The constructed features are easy to interpret using ideas from causality.
3. The similarity $s(i,j)$ seems to be novel and explicitly leverages multiple responses to decide which predictors likely share causal roles.

**Weaknesses:**

1. In causality, identifiability is one of the most important things. The author borrowed the terminology, but unfortunately can not provide any guarantees on the identifiability of the mediator and the latent variables in their method in any case (even if the ground-truth is Figure 1). Therefore, the title and the abstract are misleading. I would suggest that the author change it to be casality-inspired feature augmentation to make a clearer separation.

2. In the setting, all predictors are assumed to be non-descendants of the response variable. I wonder why it is the case and how anti-causal relationships ($Y\rightarrow X$) would affect the proposed framework. After all, if the response is the cause of a predictor $X$, then $X$ should also be included as an important feature.

3. The mediator is simply the average of the variables in the cluster. Although it might be meaningful for some data sources, it is in general a simple approximation. I wonder whether it can be extended to an arbitrary linear transformation.

4. The calculation in Table 1 is not clear to me. The probability $\pi_m,\pi_c$ for linking $X$ and $Z$ is missing. Shouldn't there also be a comparison with the probability of not having any mediator/confounder, i.e. direct link with the response? That would help to better assess the similarity metric.

5. Since the proposed method has multiple parameters to tune, a direct comparison of the runtime is necessary to assess the practicality of the method.

6. The experiments are not sufficient. For example, in the synthetic datasets, there is no test for the robustness of the method under assumption violations, e.g. alternative mediator aggregation, anticausal predictors. Only one real dataset is given, making it difficult to evaluate the performance of the method. The author could include more datasets, e.g. gene datasets like GTEx / eQTL, TCGA multi-omics, etc.

**Questions:**

See weaknesses.

---

> ### Author Response · Authors · 2025-11-21
>
> We thank the reviewer for their comments and questions. We respond to each of them below.
>
> ## W1
> We agree that the wording in the abstract (“identified causal features”) can be read as an identifiability claim. We have revised the title, abstract, and manuscript to emphasize that CFA is _causality-inspired_ feature augmentation rather than causal discovery (e.g., replacing “identified causal features” with “constructed causality-inspired features,” and clarifying that we do not claim identifiability of the true latent variables). Our causal generative model serves as a principled motivation for our similarity measure and feature construction—not as a target for exact latent-variable identification. The theoretical results (e.g., Theorem 3.1 and 3.2) are explicitly framed as guarantees about the structural relationships relevant for clustering, rather than identifiability of the latent variables themselves.
>
>
> Crucially, **CFA is not a causal discovery method.** It is a novel machine learning framework for high-dimensional regression that is the first, to the best of our knowledge, to use causal principles for feature augmentation and regression. The validity of our method does not rely on formal causal discovery or identifiability guarantees. Instead, it is justified by **(1) Theoretical soundness,** with self-contained proofs for each procedural step (e.g., Theorems 3.1 & 3.2), and **(2) External predictive utility,** where the value of our engineered features is proven by their downstream success: reducing overfitting and improving regression performance, as seen across the five real-world datasets tested in (SP500 in main text and 4 other datasets in appendix).
>
> ## W2
> We adopt the assumption that predictors are non-descendants of the responses in order to make the causal diagrams and theoretical analysis in Section 3 precise and interpretable: the generative model is written in the usual structural form “(latent) predictors → responses.” Importantly, however, the CFA algorithm itself does not require this assumption for prediction. It only relies on (i) empirical dependence tests between $X$ and $Y$, and (ii) a regularized regression of $Y$ on $([\mathbf{X}, \mathbf{Z}^{(m)}, \mathbf{Z}^{(c)}])$.
>
> If in the true data-generating process some predictors are actually effects of the responses (i.e., $Y \to X$), but they are nevertheless predictive of $Y$, CFA will still treat them as useful features. They remain in the direct-effects block $\mathbf{B}^{(d)}$, and the Elastic Net objective is free to assign them large coefficients when this improves out-of-sample performance. In other words, anti-causal predictors are not excluded or down-weighted by design; they are simply additional predictive covariates. What changes in the presence of $Y\rightarrow X$ relations is not the validity of CFA as a regression method, but the strength of the causal interpretation one can ascribe to the learned structure (mediator/confounder clusters become predictive proxies rather than literal causal parents).
>
> ## W3
> This is an interesting direction. We chose the simple cluster average as mediator primarily for robustness and interpretability: in many domains (e.g., sector returns in finance), the mean is a natural and widely used summary, and it avoids introducing additional parameters that could overfit in the low-sample regimes we target.
>
> ## W4
> Table 1 reports conditional probabilities of the form
> $$
> \mathbb{P}\bigl(|\mathbb{G}\_{ijk}|>0 ,\big|, \text{given mediator/confounder cluster type of } X_i,X_j\bigr)
> $$
> under Assumption 3.2. Because we condition on the cluster types there, only the edge probabilities $(p_{xy},p_{my},p_{cy})$ appear. The priors $\pi_m,\pi_c$ over cluster types are used later in Appendix D, where we marginalize over the five cases in Table 1 to obtain
> $$
> \mathbb{P}\bigl(|\mathbb{G}\_{ijk}|>0 \mid X_i,X_j \text{ in same cluster}\bigr)
> \quad\text{vs.}\quad
> \mathbb{P}\bigl(|\mathbb{G}\_{ijk}|>0 \mid X_i,X_j \text{ in different clusters}\bigr),
> $$
> which yields Theorem 3.2. In the revision, we will add an explicit pointer in the main text clarifying this conditioning/marginalization step.
>
> The “no mediator/confounder, only direct link” situation is already captured via $p_{xy}$. For example, in the “same mediator cluster” row, the $p_{xy}^2$ term corresponds exactly to both $X_i$ and $X_j$ connecting to $Y_k$ only through direct $X\to Y$ edges; analogous $p_{xy}$ / $p_{xy}^2$ contributions appear in the confounder rows. Appendix D shows that, in the sparse-edge regime (small $p_{xy},p_{my},p_{cy}$), the first-order terms involving $p_{my}$ and $p_{cy}$ dominate these purely direct-edge cases, which is what leads to
> $$
> \mathbb{P}\bigl(|\mathbb{G}\_{ijk}|>0 \mid X_i,X_j \text{ in same cluster}\bigr)  >  \mathbb{P}\bigl(|\mathbb{G}\_{ijk}|>0 \mid X_i,X_j \text{ in different clusters}\bigr).
> $$

---

> ### Author Response · Authors · 2025-11-21
>
> (response continued)
>
> ## W5
> We agree that practicality is important and have taken care to keep CFA’s tuning and runtime comparable to strong baselines. As noted in line 678, we constrained the regularization to be uniform across all feature types, using a single, global L1/L2 ratio and a single overall regularization strength for the combined feature set $[\mathbf{X}, \mathbf{Z}^{(m)}, \mathbf{Z}^{(c)}]$. As a result, the final regression step in our implementation of CFA has the exact same number of hyperparameters to tune as the standard Elastic Net baseline.
>
> In addition, Appendix C provides both a theoretical time-complexity analysis and empirical wall-clock measurements. In particular, Table 5 shows that even in a challenging high-dimensional setting with ($n_1 = n_2 = 4000$), our Python implementation of CFA runs in about 1.6 minutes on a MacBook Pro M1 laptop, demonstrating that the method is practically scalable. This is within a small constant factor of Elastic Net (the extra cost coming from the dependency tests and clustering).
>
> ## W6
> In addition to the S&P 500 dataset in the main text, the paper already evaluates CFA on **four further real-world multi-output regression datasets**, as stated in line 364 (“for the sake of space, we include additional experiments on four real-world datasets and true graph recovery in Appendix B”). Specifically, Appendix B reports results on:
>
> - **ATP-7D** (airline ticket prices),
> - **OES-10** (occupational employment survey),
> - **RF-2** (river flow forecasting), and
> - **SCM-1D** (supply chain management).
>
> To make this clearer, have moved a summary of the four additional real-world experiments (ATP-7D, OES-10, RF-2, SCM-1D) into the main text (new Section 4.4) and keep detailed results in the appendix.
>
> As for the synthetic experiments, they are deliberately aligned with our generative model to test recovery in the idealized case, while robustness to model misspecification is reflected in CFA’s strong performance across these diverse real-world settings.

---

> > ### Comment · Reviewer_DDDW · 2025-11-27
> >
> > Thank you for the reply. I still have 2 follow-up questions regarding W2 and W5.
> >
> > W2: I agree that anti-causal features would also be selected in general. Although there is some subtle difference in terms of constructing "mediators" on the descendant side, e.g. $Y\rightarrow M\rightarrow (X_1,X_2)$. In this case, the "mediator" is in fact the confounder of $X_1,X_2$. But I assume this can be detected by techniques in Section 3.2, as $X_1,X_2$ have high correlations. The probability on the descendant side should also be similar to the ancestor side, as described in Table 1. Therefore, I suspect that the current algorithm, **without much change**, can also deal with the case where there are mediators and confounders between/within ancestor(causal) and (anti-causal) descendant predictors. I wonder if the author could verify this rigorously. If that is the case, the modelling in this paper could become more general while using the same algorithm.
> >
> > W5: I think the claim is a bit misleading. There are 3 more hyperparameters to tune in the algorithm, and in the paper it seems like 8x more time than elastic net is needed at least. I wonder if Table 5 takes these hyperparameter tuning into account? Otherwise, is there any default hyperparameter that works well across the experiments?
> >
> > I would be willing to increase my score if my questions can be addressed.

---

> ### Author Response · Authors · 2025-12-02
>
> We thank the reviewer for engaging in the discussion. Below we provide detailed responses to their comments.
>
> ## W2 (cont.)
> We appreciate this thoughtful follow-up and fully agree with the reviewer’s intuition: CFA should also handle situations where $Y$ has descendants in $\mathbf{X}$ via latent mechanisms (e.g., $Y \to Z^{(am)} \to (X_1,X_2)$), and the current algorithm in fact already behaves that way.
>
> Algorithmically, in a pattern like $Y \to Z^{(am)} \to (X_1,X_2)$, where $am$ stands for anti-causal mediator:
> - Both $X_1$ and $X_2$ are dependent on $Y$ through the path $Y \to Z^{(am)} \to X_i$, so our similarity score $s(1,2)$ counts many shared responses and becomes large.
> - The shared latent $Z^{(am)}$ induces strong correlation among $X_1$ and $X_2$, so our intra-cluster correlation test in Sec. 3.2 classifies this group as a “confounder-type” cluster.
> - PCA on the cluster ${X_1,X_2}$ then recovers (up to scale) the dominant factor driven by $Z^{(am)}$, and this factor is used as an additional feature in exactly the same way as a confounder feature in the augmented regression.
>
> So, operationally, CFA already treats latent descendant mediators as “confounder-style” factors, exactly as the reviewer suggested.
>
> To make this more rigorous within our theoretical framework, we can extend the edge-generation model in Assumption 1 to explicitly include anti-causal mediators $Z^{(am)}$:
> - In addition to mediator nodes $\mathbf Z^{(m)}$ and confounders $\mathbf Z^{(c)}$, we allow “anti-causal mediator” nodes $Z^{(am)}\_\ell$ for each response $Y_k$.
> - For each $Y_k$, we add edges $Y_k \to Z^{(am)}\_\ell$ independently with probability $p_{ym}$, and for each predictor $X_j$ in the cluster governed by $Z^{(am)}\_\ell$, edges $Z^{(am)}\_\ell \to X_j$ with probability $p_{mx}$, all independently of the other edge-generation events.
>
> Now consider the triplet $(X_i,X_j,Y_k)$ with $X_i,X_j$ in the same anti-causal mediator cluster $Z^{(am)}\_\ell$:
> - For $X_i$ to be dependent on $Y_k$ via this mechanism, we need the edges $Y_k \to Z^{(am)}\_\ell$ and $Z^{(am)}\_\ell \to X_i$, which happen with probability on the order of $p_{ym} p_{mx}$.
> - For $X_j$ to also be dependent on $Y_k$ through the same $Z^{(am)}\_\ell$, we additionally need $Z^{(am)}\_\ell \to X_j$, which adds another factor $p_{mx}$.
>
> Thus, for same-cluster pairs, the anti-causal–mediator contribution to
> $$
> \mathbb{P}\bigl(X_i \not\perp Y_k, X_j \not\perp Y_k \mid \text{same cluster}\bigr)
> $$
> is of order $p_{ym} p_{mx}^2$, i.e., first order in the “per-cluster” parameters $p_{ym}$ and $p_{mx}$ (just as in Table 1 the dominant terms for confounder/mediator structures are first order in $p_{my},p_{cy}$).
>
> For different-cluster pairs, in the extended model a single anti-causal mediator $Z^{(am)}\_\ell$ is attached to exactly one predictor cluster. Therefore, if $X_i$ and $X_j$ belong to different clusters, they cannot share the same $Z^{(am)}\_\ell$: the event that both end up dependent on $Y_k$ through anti-causal mediators necessarily goes via two independent modules (one for each cluster), contributing terms on the order of $(p_{ym} p_{mx})^2$ to
> $$
> \mathbb{P}\bigl(X_i \not\perp Y_k, X_j \not\perp Y_k \mid \text{different clusters}\bigr),
> $$
> that is, second order in the small edge probabilities. As in our original analysis, first-order contributions arise only when $X_i$ and $X_j$ share the same latent module (mediator, confounder, or now anti-causal mediator). Hence the extended model preserves the key inequality
> $$
> \mathbb{P}\bigl(X_i \not\perp Y_k, X_j \not\perp Y_k \mid \text{same cluster}\bigr)>
> \mathbb{P}\bigl(X_i \not\perp Y_k, X_j \not\perp Y_k \mid \text{different clusters}\bigr),
> $$
> now with additional $\mathcal{O}(p_{ym} p_{mx}^2)$ terms on the left and $\mathcal{O}((p_{ym} p_{mx})^2)$ terms on the right.
>
> Finally, have made this generalization explicit in the main text. In Sec. 3.5 (“Scope of CFA”) we have added:
>
> “It is worth noting that while our derivation assumes ancestral predictors, the CFA framework generalizes to anti-causal settings where responses cause predictors via latent mechanisms (e.g., $Y \to Z^{(am)} \to \mathbf{X}$). In such cases, $Z^{(am)}$ induces correlation among $\mathbf{X}$, leading CFA to classify the group as a ‘confounder’ cluster. PCA then recovers $Z^{(am)}$ as a predictive feature for $Y$. Thus, CFA effectively captures both causal and anti-causal latent structures.”
>
> We hope this addresses the reviewer’s concern and clarifies that both the algorithm and its theoretical justification extend naturally to the richer ancestor/descendant scenario they describe.

---

> ### Author Response · Authors · 2025-12-02
>
> ## W5 (cont.)
> CFA does indeed introduce three additional structural hyperparameters beyond Elastic Net (total number of clusters $n_c + n_m$ and two $p$-value thresholds for the dependence and classification tests). What we intended to convey in the original appendix is only that, _conditional on a fixed clustering_, the final regression stage uses the **same** two regularization hyperparameters as standard Elastic Net (a single global $\ell_1/\ell_2$ ratio and a single overall penalty).
>
> ### Tuning cost
> All methods, including CFA and Elastic Net, are tuned with the same Hyperopt–TPE algorithm using a fixed budget of 50 evaluations. That is, every algorithm is _run the same number of times_; the extra structural hyperparameters in CFA increase the search dimension but not the number of runs. Table 5 reports the wall-clock time of a _single_ CFA fit (averaged over 20 random seeds) on synthetic data for varying $(n_1,n_2)$, without including Hyperopt or cross-validation for any method. We now state this explicitly.
>
> To address the concern about the impact of the additional hyperparameters on runtime, we added a new empirical study in App. C.2. On fixed synthetic data, we vary each structural hyperparameter in isolation. Varying the number of clusters $m$ from $10$ to $500$ increases the mean runtime only mildly (from $\approx 0.36$ to $\approx 0.49$ seconds), consistent with the near-linear dependence on $n_c+n_m$ predicted by our complexity analysis. In contrast, sweeping the two $p$-value thresholds over several orders of magnitude has essentially no systematic effect on runtime: the mean times stay in a narrow band, confirming that these thresholds change which edges/clusters are selected but not the asymptotic cost of the algorithm.
>
> ### New comparison with baselines
> Finally, we added a new runtime comparison with all baselines in App. C.3 on synthetic data with $N=500$ and $n_1=n_2 \in \\{200,500,1000,1500,2000\\}$ (visualized in the new runtime plot in Figure 8). This shows that CFA is actually **much faster** than Elastic Net and Lasso in the high-dimensional regime: for example, at $n_1=n_2=2000$, CFA fits in roughly $8$ seconds, whereas Elastic Net and Lasso take about $40$–$50$ seconds (a $5\times$–$6\times$ slowdown). Intuitively, the extra structure-learning stage pays off computationally: after clustering, the augmented features $[\mathbf{X},\mathbf{Z}^{(m)},\mathbf{Z}^{(c)}]$ concentrate shared variation into a smaller number of information-rich directions, so the downstream Elastic Net inside CFA sees a much simpler, better-conditioned optimization problem and converges in far fewer iterations than plain Elastic Net on the raw $\mathbf{X}$.

---

### Author Response · Authors · 2025-11-21

### We thank all reviewers for their careful reading of our paper and for the constructive feedback. We have revised the manuscript accordingly and highlighted all changes in blue in the updated version. Below, we respond in detail to each comment and question raised by the reviewers, organized by reviewer and point, and indicate how and where the manuscript has been updated in response.

---

### Author Response · Authors · 2025-12-02

Dear Area Chair,

We understand the unique circumstances of this review process. To assist in your assessment, we provide a concise summary of our rebuttal and the revisions made to the manuscript (changes marked in blue in the PDF).

---

### 1. Scope Clarified: Feature Augmentation, Not Causal Discovery

Several criticisms (esp. Reviewer DDDW) judged the paper as if it were trying to *identify* the true causal graph and latent variables.

- We explicitly clarify that CFA is **not** a causal discovery method, but a regression framework that uses causal motifs (mediators/confounders) as *design principles* for augmenting the regression problem with better features.
- Title, abstract, intro, and a dedicated “Scope of CFA” subsection now emphasize that we **do not claim identifiability**; the theory is about our similarity metric and clustering, not recovering the DAG.
- We also make explicit that the same mechanism works in anti-causal settings ($Y \to X$), as pointed out by Reviewer DDDW: latent descendant factors (e.g. $Y \to Z^{(am)} \to X$) are automatically treated as “confounder-type” clusters, and PCA recovers predictive factors that are used exactly as in the causal case.

---

### 2. Experiments: 5 Real-World Datasets, Not Just S\&P 500

Reviewers incorrectly described the experiments as weak or “only S\&P 500,” overlooking the four real-world datasets already in the appendix.

- We now summarize all 5 real-world multi-output low-sample datasets in the main text: S\&P 500, ATP-7D, OES-10, RF-2, SCM-1D.
- Across all five, CFA consistently achieves the **best test performance**, often with smaller train–test gaps than strong baselines (ElasticNet, Ridge, Lasso, S-VAE, PCR, PLS).
- On S\&P 500, absolute correlations are small (as is standard in next-day return prediction), but CFA consistently improves over baselines in every rolling window.

---

### 3. Efficiency and Practicality: New Runtime Benchmarks

There were concerns that our similarity/clustering step would be too slow.

- We added **new runtime experiments** comparing CFA against all baselines, given in Appendix C.2 and C.3.
- In the high-dimensional regime (e.g. $N = 500$, $n_1 = n_2 = 2000$), CFA is about **5–6x faster than ElasticNet and Lasso** in wall-clock time.
- Intuition: CFA spends a small upfront cost to build information-rich latent features; the downstream Elastic Net then solves a much better-conditioned problem and converges substantially faster than ElasticNet on just $\mathbf{X}$.

---

### 4. Literature and Novelty

We briefly strengthened and clarified the positioning:

- We added recent works (e.g., Xu et al. 2019, Bertsimas \& Van Parys 2017) and emphasize that CFA targets dense, high-dimensional, low-sample, multi-output regression, where methods like SHORE (sparse outputs) and group/multi-task lasso (shared support across tasks) are misaligned and empirically underperform in our benchmarks.
- Algorithmically, our paper introduces:
  - a **new similarity metric** based on shared $Y$-dependencies with theoretical justification;
  - a **mediator vs confounder clustering + feature construction** pipeline; and
  - an **augmented regression scheme** that plugs into standard Elastic Net and consistently improves sample efficiency.
  - an **open-source Python implementation** with a scikit-learn–style regressor API, making CFA easy to drop into existing workflows while remaining very fast and performant in practice.

---

We thank the Area Chair for their time and effort in reviewing our work and rebuttal under these exceptional circumstances.

Best regards,

The Authors

---

### Meta-Review · Area_Chair_kts6 · 2026-01-01

**Summary:**

The reviewers’ main concerns focus on the strength and justification of the modeling assumptions, the limited scope and sufficiency of the experimental evaluation, and the unclear practical impact of the proposed method. In particular, several reviewers question whether the restrictive linear and special graph structure assumptions are realistic or broadly applicable, and whether the method remains valid under model mismatch or more complex graph structures. Reviewers also note that the empirical evaluation is limited, relies heavily on synthetic or narrowly scoped real-world datasets, and lacks sufficient comparisons with relevant modern baselines. Finally, the practical usefulness of the method and the scenarios in which it provides clear advantages remain insufficiently demonstrated.

**Reviewer Concerns:**

Main concerns raised by the reviewers: concerns about the assumptions underlying the proposed causal model, and concerns about the sufficiency of the experimental evaluation.

----

As a primary concern, all reviewers question the assumptions required by the causal analysis. For example, Reviewer DDDW raises concerns about the anti-causal structure, Reviewer 4Jaf points out potential model mismatch, and Reviewer eEeh notes that the true shared variability may involve multiple factors. While it is generally necessary to introduce assumptions in causal analysis, in this work the function class is already restricted to linear models. In this context, introducing several additional strong assumptions further limits the generality of the approach and weakens the universality of the claims. Moreover, such assumptions are often difficult to justify in practice.

In the rebuttal, the authors provide some initial discussion (e.g., an analysis of the anti-causal case). However, this analysis is not sufficiently rigorous to fully address the concerns. Given the linear setting, stronger theoretical results could reasonably be expected, and supporting simulation experiments to illustrate when the assumptions hold or fail would be relatively straightforward to implement. These would be necessary to convincingly justify the modeling choices.

Even setting aside the assumptions, the main unresolved concerns then shift to the experimental evaluation, particularly in relation to modern methods and realistic settings. In the revised version, the authors add references to some recent works (e.g., Xu et al., 2019; Bertsimas & Van Parys, 2017), but this is not sufficient to address the reviewers’ requests. For example, in the context of linear causal models, there has been recent progress in causal discovery with latent variables, which would be reasonable baselines for comparison but are currently missing. In addition, recent foundation models for time series (considering the used datasets in this work) could also be considered as relevant points of reference.

Overall, while the rebuttal and revision clarify certain aspects and add some references, the core concerns regarding the strength and justification of the assumptions, as well as the adequacy and breadth of the experimental evaluation, remain outstanding.

**Reviewer Scores:**

Reviewer DDDW:

Although the authors provided a detailed and initial response to the anti-causal and runtime questions, the additional material remains largely conceptual and does not include new formal theoretical guarantees or new empirical validations of robustness under assumption violations. As such, while the discussion improves clarity, it is unlikely to fully resolve the reviewer’s concerns or lead to a score increase.

----

Reviewer 4Jaf:

The reviewer’s core concerns regarding limited novelty, insufficiently convincing experiments, and missing comparisons with modern alternatives are only partially addressed. A score change is therefore unlikely.

---

Reviewer eEeh:

The reviewer’s main concerns about practical usefulness, missing baselines, and limited exploration of multi-factor latent structure remain largely outstanding.

---

Overall: The rebuttal improves clarity and addresses some technical points, but does not substantially change the balance of evidence with respect to the main concerns. Therefore, no positive score changes are expected from any reviewer.

---

### Decision · Program_Chairs · 2026-01-26

Reject